



# Quantifying sources of Brazil's CH$_4$ emissions between 2010 and 2018 from satellite data

Rachel L. Tunnicliffe[1,2], Anita L. Ganesan[1], Robert J. Parker[3,4], Hartmut Boesch[3,4], Nicola Gedney[5], Benjamin Poulter[6], Zhen Zhang[7], Jošt V. Lavrič[8], David Walter[8,9], Matthew Rigby[2], Stephan Henne[10], Dickon Young[2], and Simon O'Doherty[2]

[1]School of Geographical Sciences, University of Bristol, Bristol, UK
[2]School of Chemistry, University of Bristol, Bristol, UK
[3]National Centre for Earth Observation, University of Leicester, Leicester, UK
[4]Earth Observation Science, School of Physics and Astronomy, University of Leicester, Leicester, UK
[5]Met Office Hadley Centre, Joint Centre for Hydrometeorological Research, Exeter, UK
[6]NASA Goddard Space Flight Center, Biospheric Sciences Laboratory, Greenbelt, USA
[7]Department of Geographical Sciences, University of Maryland, College Park, USA
[8]Max Planck Institute for Biogeochemistry, Mainz, Germany
[9]Max Planck Institute for Chemistry, Mainz, Germany
[10]Empa, Swiss Federal Laboratories for Materials Science and Technology, Dübendorf, Switzerland

**Correspondence:** Rachel Tunnicliffe (rachel.tunnicliffe@bristol.ac.uk)

**Abstract.** Brazil's CH$_4$ emissions over the period 2010–2018 were derived for the three main sectors of activity: anthropogenic, wetland and biomass burning. Our inverse modelling estimates were derived from GOSAT satellite measurements of XCH$_4$ combined with surface data from Ragged Point, Barbados and the high-resolution regional atmospheric transport model NAME. We find that Brazil's mean emissions over 2010–2018 are $33.6 \pm 3.6 \, \mathrm{Tg \, yr^{-1}}$, which are comprised of
5  $19.0 \pm 2.6 \, \mathrm{Tg \, yr^{-1}}$ from anthropogenic (primarily related to agriculture and waste), $13.0 \pm 1.9 \, \mathrm{Tg \, yr^{-1}}$ from wetlands and $1.7 \pm 0.3 \, \mathrm{Tg \, yr^{-1}}$ from biomass burning sources. In addition, between the 2011–2013 and 2014–2018 periods, Brazil's mean emissions rose by $6.9 \pm 5.3 \, \mathrm{Tg \, yr^{-1}}$ and this increase may have contributed to the accelerated global methane growth rate observed during the latter period. We find that wetland emissions from the Western Amazon increased during the start of the 2015–16 El Niño by $3.7 \pm 2.7 \, \mathrm{Tg \, yr^{-1}}$ and this is likely driven by increased surface temperatures. We also find that our
10  estimates of anthropogenic emissions are consistent with those reported by Brazil to the United Framework Convention on Climate Change. We show that satellite data is beneficial for constraining national-scale CH$_4$ emissions, and, through a series of sensitivity studies and validation experiments using data not assimilated in the inversion, we demonstrate that calibrated ground-based data are important to include alongside satellite data in a regional inversion, and that inversions must account for any offsets between the two data streams and their representations by models.



## 1 Introduction

Methane ($CH_4$) is the second most important anthropogenic greenhouse gas behind carbon dioxide due to its radiative properties and atmospheric abundance (Ciais et al., 2013). After a brief plateau period around the turn of the century (Cunnold, 2002; Dlugokencky et al., 2003), $CH_4$ mole fractions began rising again globally after 2007 (Rigby et al., 2008; Dlugokencky et al., 2009; Frankenberg et al., 2011; Nisbet et al., 2016) with some of the strongest growth rates occurring from 2014 onward (Nisbet et al., 2019). This increase in $CH_4$ growth rate was accompanied by a shift in the $\delta^{13}$C-$CH_4$ isotopic ratios to more negative values, suggesting a change in the global makeup of sources and/or sinks. The drivers responsible for this shift are presently not well-understood, and proposals include increases from tropical wetlands or agriculture, decreases in biomass burning, changes in fossil fuel emissions or in the hydroxyl radical sink (e.g. Monteil et al. 2011; Schaefer et al. 2016; Schwietzke et al. 2016; Nisbet et al. 2016; Rigby et al. 2017; Worden et al. 2017; McNorton et al. 2018; Turner et al. 2019; Nisbet et al. 2019). Quantifying the $CH_4$ budget and understanding how major sources and sinks have evolved is key to designing emission pathways that limit global warming due to the importance of $CH_4$ in meeting global climate targets (Ganesan et al., 2019; Nisbet et al., 2019, 2020).

The Paris Agreement pledges to limit warming to less than 2°C with an aspiration for less than 1.5°C warming from pre-industrial levels (UNFCCC, 2015). The mitigation action taken by each country is dependent on their own Nationally Determined Contributions and accounting for national emissions will occur through inventory or "bottom-up" methods. To assess whether these self-determined targets are being met, independent estimates can be derived using "top-down" strategies that use atmospheric measurements to quantify sector-level emissions estimates at near real-time and at high-resolution (e.g. Ganesan et al., 2019). Using both top-down and bottom-up methods together for national-scale greenhouse gas estimation is considered to be best practice (Calvo Buendia et al., 2019) and allows for the greatest process-level understanding of changes in the atmosphere.

Brazil is thought to be a major contributor to global $CH_4$ emissions due to its variety of natural and human-made sources. Anthropogenic emissions arise from agriculture, waste and biomass burning (Ministry of Foreign Affairs et al., 2019). Brazil's 2018 Biennial Update Report to the United Framework Convention on Climate Change (UNFCCC) states that $17.6 \pm 4.0$ Tg of $CH_4$ was emitted from anthropogenic sources in 2015. The majority of these emissions were from agricultural processes (70% from enteric fermentation, manure management and crop residue burning) with the remainder coming from waste (16%), energy (4%) and land-use change (6%) (Ministry of Foreign Affairs et al., 2019).

Around 60% of the Amazon basin and 80% of the Pantanal wetland region (Ministry of Science and Innovation, 2016; Schulz et al., 2019) exist within Brazil in the northern and central-western regions of the country, respectively. The primary areas of agricultural activity are in central and southern provinces and include cattle ranching and sugar cane production, while waste and fossil fuel emissions are focused in population centres along the eastern coast (Ministry of Foreign Affairs et al., 2019). Biomass burning occurs along the 'arc of deforestation' along the southern edge of the Amazon rain forest during and after the dry season (Jul-Oct). This is in contrast to Amazon wetland emissions which peak during and after the wet season (Dec-Mar).



Current top-down estimates of $CH_4$ emissions from Brazil, the Amazon and tropical South America vary depending on the method, source of data and area considered. Across South America's Amazon basin, total emission estimates derived from aircraft measurements for 2010–2013 are between $31–43\,\mathrm{Tg\,yr^{-1}}$ (Wilson et al., 2016; Pangala et al., 2017). A recent study that used regional inversions with satellite data by Janardanan et al. (2019) found Brazil's emissions, on average, to be $56.2\,\mathrm{Tg\,yr^{-1}}$ from 2011–2017. Many previous studies have estimated emissions globally using satellite data (e.g., Bergamaschi et al. 2009; Feng et al. 2017). In the synthesis of Saunois et al. (2016), across the Tropical South America region, wetland emission estimates derived using different datasets and top-down methods span the large range of $23–64\,\mathrm{Tg\,yr^{-1}}$. The wide range of estimates indicate that large uncertainties exist and these uncertainties are exacerbated when estimating emissions over smaller scales such as the Amazon basin or when quantifying individual sources.

Through use of a high-resolution regional inversion framework coupled with satellite measurements of $CH_4$, we inferred spatial and temporal distributions of Brazil's $CH_4$ emissions from 2010–2018. The regional inversion approach provides the benefit that uncertainties in the hydroxyl radical $CH_4$ sink (Rigby et al., 2017; Turner et al., 2017; Nguyen et al., 2020), a limitation in global approaches, can be neglected. Owing to a spatial and temporal difference in Brazil's major $CH_4$ sources, these emissions are further partitioned into source sectors (Section 3.1) and are also presented for different wetland regions (Section 3.2). We demonstrate the importance of the inversion setup when using satellite data to estimate country and basin-scale emissions. Independent validation using *in-situ* data is shown in Section 3.3 and sensitivity studies, testing a range of different input factors, are discussed in Section 3.4.

## 2 Methods

### 2.1 $CH_4$ measurements

We used data from three sources: (1) University of Leicester v7.2 total column $CH_4$ product from the Thermal And Near-infrared Sensor for carbon Observation Fourier Transform Spectrometer (TANSO-FTS) instrument on board the Greenhouse gases Observing SATellite (GOSAT) from April 2010 - Nov 2018, (2) surface data from Ragged Point, Barbados (RPB, $+13.17°\mathrm{N}$, $-59.43°\mathrm{E}$), and (3) surface data from the Amazon Tall Tower Observatory (ATTO, $-2.15°\mathrm{N}$, $-59.01°\mathrm{E}$) for external validation of the inversion. Figure 1 shows the positions of GOSAT points for a one year period and the locations of the two surface stations.

Dry-air column-averaged $CH_4$ mole fractions ($XCH_4$) were derived using the $CO_2$ proxy method, which multiplies the $XCH_4/XCO_2$ ratio by a model $XCO_2$ field (Parker et al., 2011, 2015). The model $XCO_2$ is based on the median of three global models which all assimilated surface site measurement data: GEOS-Chem (Feng et al., 2011), Carbon Tracker (Peters et al., 2007) and LMDZ (MACC/CAMS) (Chevallier et al., 2010). This GOSAT product was previously compared to aircraft measurements over the Amazon basin by extrapolating the aircraft profiles through the troposphere and using a stratospheric model, and showed differences that ranged from $-1.9–9.7\,\mathrm{nmol\,mol^{-1}}$ (Webb et al., 2016).

We used Level 2 GOSAT measurements that were taken in nadir mode within an area that extended from $-35.8$ to $7.3°\mathrm{N}$ and $-76.0$ to $-32.8°\mathrm{E}$ and that passed the quality threshold. We only used nadir measurements to minimise the effect of any



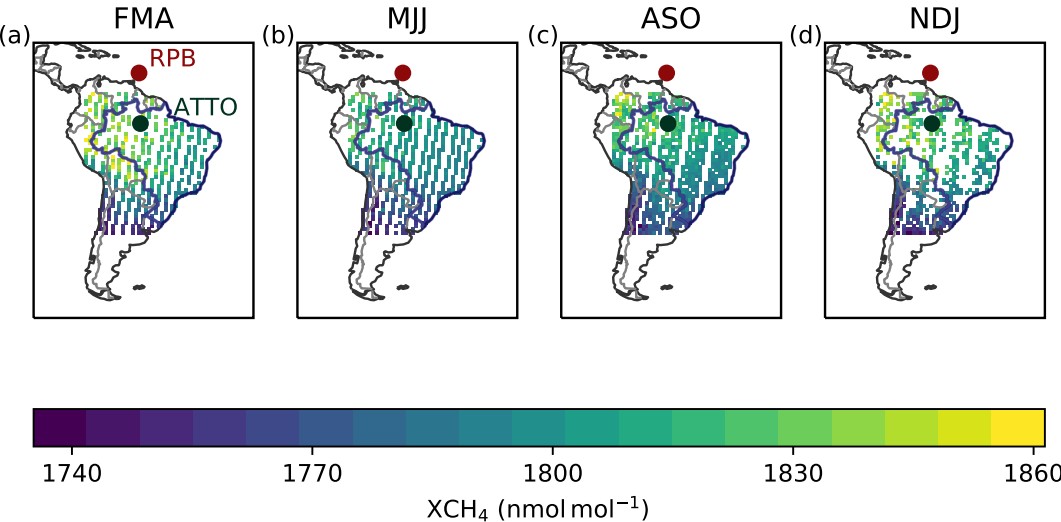

**Figure 1.** GOSAT measurements grouped into February-April (FMA), May-July (MJJ), August-October (ASO) and November-January (NDJ). Measurements are shown here for the 02/2014–01/2015 period and averaged in $1°$ bins for clarity. FMA and ASO are representative of the wet and dry seasons over the Amazon, respectively. Positions for the ATTO and RPB sites are also shown. The Brazilian border is highlighted in blue.

unquantified biases between nadir and glint mode. In addition, we filtered data where the surface pressure deviated from the retrieval grid by enough to reduce the number of retrieval levels to less than 20. The remaining data points were averaged

across a $0.23 \times 0.35°$ grid to match the lowest resolution of the atmospheric transport model grid cell (see section 2.3) across the 2010–2018 time period leaving $\sim 1300$ data points on average per month.

Data from RPB were used alongside the satellite measurements in the inversion to provide additional constraints on the boundary conditions. RPB is part of the Advanced Global Atmospheric Gases Experiment (AGAGE) network (Prinn et al., 2018) and predominantly measures well-mixed background air. Measurements up to 2017 were made using GC-FID (Gas

Chromatography Flame - Ionization Detector) and beyond this with a CRDS (Cavity Ring-down Spectrometer) instrument. All data were averaged into hourly samples.

Measurements from ATTO (Andreae et al., 2015; Botía B. et al., 2019) were used for external validation of the inversion results. ATTO is located near Manaus within the Amazon rainforest. The position and predominant north-easterly wind direction means that this site is particularly sensitive to $CH_4$ emitted from wetlands but may also receive air masses from regions

of biomass burning and other human activity (Andreae et al., 2015; Pöhlker et al., 2019). $CH_4$ mole fractions from 2014–2018 derived from CRDS instrumentation have been used in this study. Hourly mean measurements from the highest inlet on the tower, at $79\,\mathrm{m}$, were used as they are assumed to be the most representative of regional air masses.





## 2.2 Atmospheric Transport Model

To provide the relationship between atmospheric mole fractions at a receptor and a surface emissions field, we used the
high-resolution Lagrangian atmospheric transport model NAME (Numerical Atmospheric dispersion Modelling Environment)
(Jones et al., 2007). Model particles were released for each GOSAT and surface measurement time and location and tracked
backward in time for 30 days. The model tracked the interaction of these particles with the surface (defined as $0–40\,\mathrm{m}$ above
modelled ground level) to quantify the sensitivity to regional emissions. The times and locations that particles left the model
domain was recorded to quantify the sensitivity to boundary conditions. NAME was driven by meteorological inputs from the
Unified Model (UM) spanning resolutions between $0.23$ to $0.09°$ latitude and $0.35$ to $0.14°$ longitude over the 2010–2018
period. The annual mean sensitivity to GOSAT measurements used in this study are shown in Appendix Fig. A1.

Satellite measurements require footprints of the total atmospheric column and model particles were released at multiple
heights based on the pressure levels defined within the GOSAT product (see Ganesan et al. 2017 for a description of how
NAME was used to simulate $XCH_4$ by applying averaging kernels, pressure weights and *a priori* information for satellite
data). The main modification in the NAME setup from Ganesan et al. (2017) made here is that surface pressure in GOSAT was
corrected to match the surface pressure from the UM. Occasionally, the corrected surface pressure level was lower than the
first model level, and in these cases, the retrievals were discarded. This ensured consistency between the model defining the
GOSAT pressure levels and NAME.

## 2.3 Inversion method

Top-down emissions estimates were inferred using a hierarchical Bayesian inversion method with reversible jump, trans-
dimensional Markov chain Monte Carlo (MCMC). A full description of the method can be found in Ganesan et al. (2014) and
Lunt et al. (2016). The hierarchical component employs a set of hyperparameters that define the model-measurement and prior
emissions uncertainties, and which were explored as part of the inversion. Inclusion of these additional model parameters allows
for uncertainties in the system to be more accurately captured. The trans-dimensional component of the inversion allowed for
the spatial inversion grid to be estimated as part of the inversion, rather than being defined *a priori*.

The *a priori* inputs to the inversion are described in Section 2.4. The emissions PDF was defined as lognormal to prevent
non-physical negative solutions from being reached. The standard deviation of this PDF was allowed to vary between 0.05 and
20.0 (with a value of one being equivalent to the prior emissions magnitude). The model-measurement uncertainty was defined
with a uniform distribution ranging from 0.2 to $200\,\mathrm{nmol\,mol^{-1}}$.

Each month, we estimated emissions from within the NAME domain (at the resolution explored by the trans-dimensional
method), as well as offsets to *a priori* boundary condition "curtains" on each edge of the domain (Section 2.4). In addition, an
offset parameter was included to account for any differences between the satellite and the calibrated ground-based measure-
ments and their representation by models. The necessity of this parameter to produce the most robust results is discussed in
Section 3.3.





**Table 1.** Summary of *a priori* emissions and boundary conditions used in this study. All maps have been re-gridded to $0.23°$ latitude by $0.35°$ longitude resolution. * Repeats this year thereafter. † For 2018, a climatological mean of 2011–2017 period was used.

| Field | Source | Resolution | Time Period | Modifications |
|---|---|---|---|---|
| Anthropogenic emissions | EDGAR v4.3.2 | Annual | 2010–2012* | Excluded agricultural waste burning and combustion from manufacturing, solid waste and fossil fuels. |
| Wetland emissions | JULES / SWAMPS | Monthly | 2010–2017* | Emissions from JULES over fractional wetland extent from SWAMPS. Total wetland emissions scaled to $44\,\mathrm{Tg\,yr^{-1}}$ |
| Biomass burning emissions | GFED v4.1 | Monthly | 2010–2015* | *None* |
| $CH_4$ mole fraction curtains | CAMS v17r1 | Monthly | 2010-2017$^†$ | *None* |

The Metropolis-Hastings MCMC sampler was run with $500,000$ iterations with the initial $100,000$ samples discarded as burn-in. Every $500^{th}$ iteration was saved and used to build posterior PDFs for each parameter. The mean and 2.5–97.5 percentiles were used to produce posterior estimates and $95\%$ confidence intervals.

## 2.4   *A priori* fields

*A priori* emissions and boundary condition fields are summarised in Table 1. Emissions were inferred for the three major source
sectors in Brazil: anthropogenic, biomass burning and wetlands. Maps for two representative months in the wet (January) and dry (September) seasons for 2014 are shown for each sector in Fig. 2.

Anthropogenic emissions, excluding biomass burning, were from the EDGAR (Emission Database for Global Atmospheric Research) v4.3.2 database (Janssens-Maenhout et al., 2017). Annual emissions were available up to 2012 and then assumed to be equal to the 2012 emissions thereafter. The biomass burning contribution was from GFED (Global Fire Emissions Database)
v4.1 (Van Der Werf et al., 2017) at monthly resolution to the year 2015 and assumed to be held at 2015 values thereafter.

Wetland emissions were based on the output from the JULES land surface model (Clark et al., 2011) which was modified to use the wetland fractional map from Surface WAter Microwave Product Series (SWAMPS). We used a version of SWAMPS that was updated from Schroeder et al. (2015) to include wetlands occurring under dense canopies, to remove rice agriculture and to include any inland water. Wetland emissions across South America were scaled to $44\,\mathrm{Tg\,yr^{-1}}$ based on the mean
bottom-up estimate for Tropical South America from Saunois et al. (2016).

*A priori* mole fractions at the boundaries of the domain were derived from the CAMS $CH_4$ flux inversion product v17r1 (accessible at https://apps.ecmwf.int/datasets/data/cams-ghg-inversions/). This version assimilated the global surface measurement network and did not use satellite data. This product was only available up to 2017, so to extend the analysis to 2018, the climatological mean of the 2010–2017 period was used.

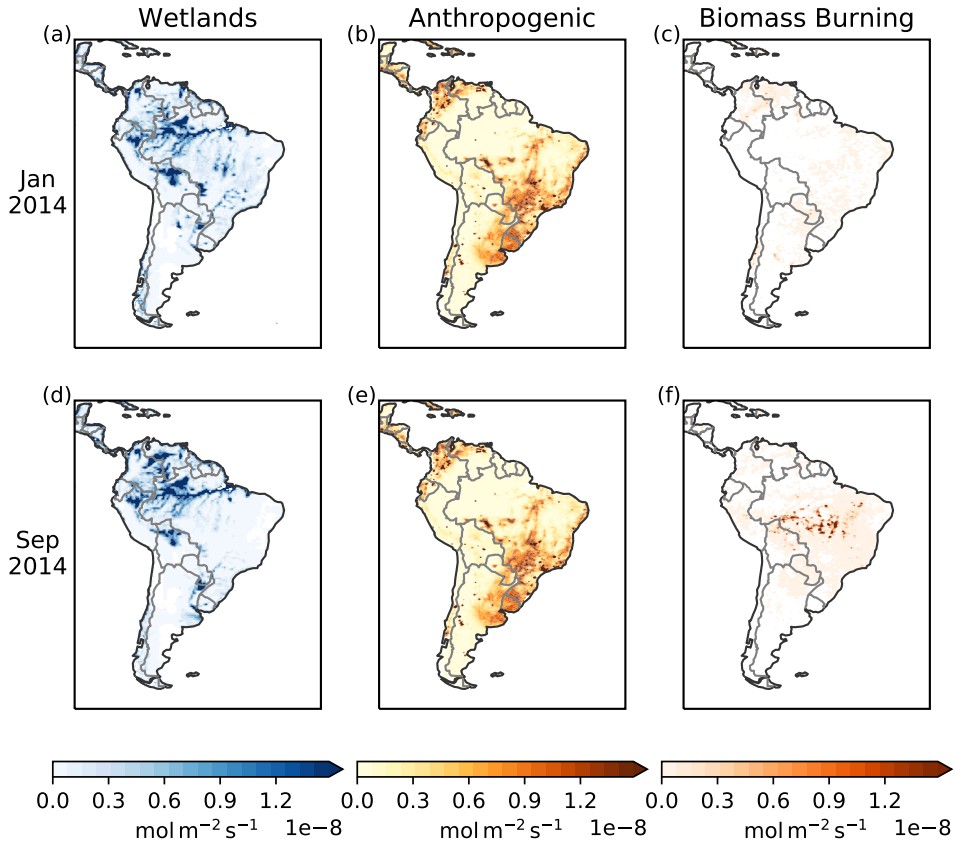

**Figure 2.** *A priori* emissions for wetlands, anthropogenic and biomass burning sectors for January and September, 2014. These months are representative of peak wetland extent in the wet season (January) and biomass burning activity (September). Note that the EDGAR anthropogenic inventory is annual resolution and is shown here for 2012.

## 2.5 Sector attribution

The total emissions estimated from the inversion were partitioned into each of the three major source sectors using the fraction of each source in the *a priori* emission fields in each grid cell. Due to the largely distinct spatial or temporal distributions of the sectors as shown in Fig. 3, the fractional map of each source is not overly dependent on the inventories used. The influence of the *a priori* distributions on the robustness of the sector partitioning is discussed in Section 3.4.

## 2.6 Validation with ATTO

To provide a validation of the inversion results, we compared a model prediction of mole fractions at ATTO derived from hourly NAME sensitivities convolved with our posterior emissions maps and boundary conditions against measured values. Four tests

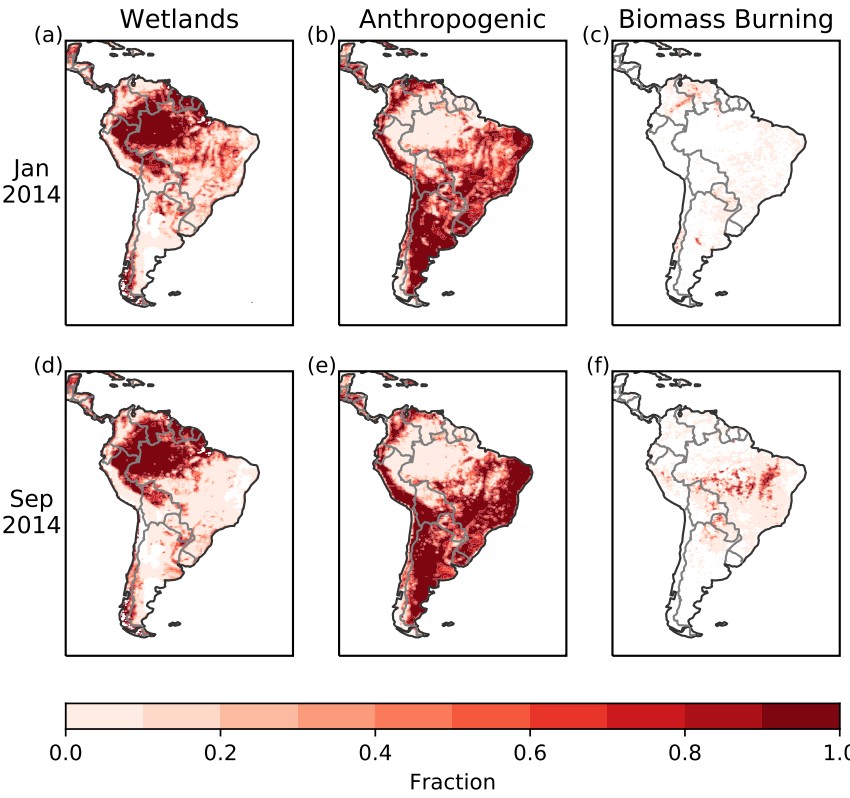

**Figure 3.** Fraction of wetland, anthropogenic and biomass burning emissions for January (top panel) and September (bottom panel) 2014 derived from the *a priori* emissions. Note that the EDGAR anthropogenic inventory is annual resolution and is shown here for 2012.

were run using different configurations of the inversion. The first three estimates were from inversions that used variants of the GOSAT and RPB dataset. The first inversion utilised GOSAT data alone. The second inversion used both GOSAT and RPB measurements but did not include an offset parameter between satellite and surface data in the inversion. The third inversion used GOSAT and RPB measurements and included an offset parameter that was estimated in the inversion (our main results). These tests and the resulting comparisons with ATTO data allowed us to determine the factors that are most important when using satellite data to constrain country-scale emissions. We performed a final test which scaled our posterior emissions map so that emissions from the Brazilian Amazon matched those derived by Wilson et al. (2016) using four aircraft sites. For the whole Amazon Basin, the lowest value in the range presented in Wilson et al. (2016) of $31.6\,\mathrm{Tg\,yr^{-1}}$ was used, with $\sim 19\,\mathrm{Tg\,yr^{-1}}$ coming from the Brazilian Amazon, based on wetland extent. This test allowed us to investigate the fit of previous results against ATTO data.





In addition to these experiments, we simulated the model prediction at ATTO using a second regional Lagrangian model, the FLEXible PARTicle dispersion model (FLEXPART), for 2014–2017 (Pisso et al., 2019). The setup for FLEXPART was the same as NAME, except the surface was defined as $0$–$50\,\mathrm{m}$ above ground level and the meteorological drivers were $1°$ resolution from the European Centre for Medium-Range Weather Forecasts (ECMWF). Particles were tracked backwards for 30 days. This test allowed us to assess whether results are significantly impacted by systematic uncertainties in NAME.

### 2.7 Sensitivity Studies

Sensitivity tests against a range of inputs to the inversion were performed to assess the robustness of our results. Three categories of inputs were tested: *a priori* emissions, *a priori* boundary conditions and the model $XCO_2$ fields used to derive $XCH_4$. In most cases, comparisons were performed for 2014 only, but if differences were seen, the analysis was expanded across the entire time range of 2010–2018. The sensitivity study details are summarised in Table 2.

To test the sensitivity to *a priori* emissions, we ran a set of inversions where emissions were perturbed one at a time from each source sector. We changed the magnitudes of emissions from each sector and tested variations of wetland extent maps. For the latter, three additional wetland distributions were used: two using JULES emissions either with Bergamaschi et al. (2007) (hereafter referred to as Kaplan, which is based on land cover maps from optical imagery) or the high-resolution Tropical and Sub-Tropical Wetland Distribution v2.0 (Gumbricht et al. 2017, hereafter referred to as Gumbricht). The final variation used the Wetland $CH_4$ emission and uncertainty dataset for atmospheric chemistry and transport modelling (WetCHARTs) based on an ensemble of wetland models (Bloom et al., 2017). We did not modify these wetland distributions to include any emissions that might occur when the water table is below the surface. These four wetland distribution maps are shown in Appendix Fig. A2.

To test the sensitivity to *a priori* boundary conditions, we used a variation of the global mole fractions used to generate the boundary condition curtains. We used the climatological mean of the MOZART global model (Emmons et al., 2010) over 2010–2014 time period. The setup for MOZART is described in Palmer et al. (2018).

To test the sensitivity to the model $XCO_2$ used to derive $XCH_4$, we generated 10 variations of $XCH_4$ for each measurement. These were created by randomly selecting between the median (the main results) and the extremes in the ensemble members that are included with the data product. We re-ran the inversion for each of the ten datasets for the full 2010–2018 time period, which allowed us to investigate random errors in $XCO_2$.

## 3 Results

### 3.1 Annual and seasonal emissions by sector

Mean emissions from 2010–2018 for Brazil are $33.6\pm3.6\,\mathrm{Tg\,yr^{-1}}$ (Fig. 4). These emissions correspond to mean anthropogenic emissions of $19.0\pm2.6\,\mathrm{Tg\,yr^{-1}}$, mean wetland emissions of $13.0\pm1.9\,\mathrm{Tg\,yr^{-1}}$, and mean biomass burning emissions of





**Table 2.** Sensitivity studies performed in this study for *a priori* fields or for model $XCO_2$. For the *a priori* emissions, each sector was varied one at a time over the full South America domain with other sectors kept in their original configurations.

† Based on bottom-up emissions estimates from Saunois et al. (2016) for tropical South America.

| Experiment category | Experiment name | Description |
|---|---|---|
| Wetland distribution | Kaplan | JULES emissions and wetland extent from Bergamaschi et al. (2007). Wetlands emissions scaled to $44\,\mathrm{Tg\,yr^{-1}}$. |
| | Gumbricht | JULES emissions and wetland extent from Gumbricht et al. (2017). Wetlands emissions scaled to $44\,\mathrm{Tg\,yr^{-1}}$. |
| | WetCHARTs | Wetland $CH_4$ emissions from WetCHARTs v1.0 (mean of extended model ensemble) (Bloom et al., 2017). |
| Wetland magnitude | Saunois high | Wetlands emissions (JULES emissions and SWAMPS extent) scaled to $34\,\mathrm{Tg\,yr^{-1}}$ †. |
| | Saunois low | Wetlands emissions (JULES emissions and SWAMPS extent) scaled to $50\,\mathrm{Tg\,yr^{-1}}$ †. |
| Anthropogenic magnitude | EDGAR x 2.0 | Anthropogenic emissions (EDGAR v4.3.2) doubled to $77.5\,\mathrm{Tg\,yr^{-1}}$. |
| Biomass burning magnitude | GFED x 2.0 | Monthly biomass burning emissions (GFED v4.1) doubled (monthly max. $20.9\,\mathrm{Tg\,yr^{-1}}$). |
| Boundary conditions | MOZART | Climatological mean of MOZART model (Emmons et al., 2010; Palmer et al., 2018). |
| $XCO_2$ variations | $XCH_4$ samples | 10 $XCH_4$ datasets created by randomly selecting across the median and the extremes for the $XCO_2$ model ensemble, compiled of GEOS-Chem, CarbonTracker and LMDZ (MACC/CAMS). |


$1.7 \pm 0.3\,\mathrm{Tg\,yr^{-1}}$. Maps of these posterior emissions and the difference from the *a priori* inputs are shown for each season in Appendix Fig. A3.

Both our 2012 and 2015 estimates of anthropogenic emissions of $16.2 \pm 3.0\,\mathrm{Tg\,yr^{-1}}$ and $18.3 \pm 2.5\,\mathrm{Tg\,yr^{-1}}$ are consistent within uncertainties with Brazil's Third Biennial Update Report to the UNFCCC, which estimates $15.6\,\mathrm{Tg\,yr^{-1}}$ in 2012 and $16.3 \pm 3.8\,\mathrm{Tg\,yr^{-1}}$ in 2015 (Ministry of Foreign Affairs et al., 2019), when LULUCF contributions are removed.

The overall rise in emissions over the 2010–2018 period generally occurred in late 2013 and early 2014 and was sustained thereafter. Average emissions between 2014–2018 rose over 2011–2013 levels by $6.9 \pm 5.3\,\mathrm{Tg\,yr^{-1}}$ and this is driven

by changes in anthropogenic, wetland, and biomass burning emissions of $3.3 \pm 3.7\,\mathrm{Tg\,yr^{-1}}$, $2.6 \pm 2.8\,\mathrm{Tg\,yr^{-1}}$, and $1.0 \pm 0.4\,\mathrm{Tg\,yr^{-1}}$, respectively.

Across 2010–2018, we find that total emissions maximise in April and minimise in October, and the overall seasonality reflects the net effect of different seasonal patterns in the three sectors. Anthropogenic emissions, the largest sector, peak in April and are lowest in August - October (dry season), and could be a result of seasonality in cattle, manure management

(e.g. Cardoso et al. 2019) or landfill emissions (e.g. Machado et al. 2009; Imbiriba et al. 2020). Anthropogenic emissions are only estimated annually in EDGAR and in reports to the UNFCCC, and thus do not capture this important feature. Wetland emissions peak during the wet season between February and April and are lowest in October and this seasonality is more pronounced in our estimates than in the *a priori* emissions. Anthropogenic and wetland emissions are discussed further for different regions of Brazil in Section 3.2. Biomass burning emissions maximise in September and the seasonality is consistent

with GFED.

Our analysis shows that individual years exhibit features that are not present in the bottom-up estimates. We find the largest biomass burning emissions in 2010, a year with strong drought and intensive burning due to high Atlantic sea surface temperatures (Lewis et al., 2011; van der Laan-Luijkx et al., 2015); annual mean emissions in 2010 were $5.5 \pm 0.5\,\mathrm{Tg\,yr^{-1}}$ (based on Apr-Dec due to availability of GOSAT measurements in 2010), but with a monthly value in September at $22.3^{+2.6}_{-2.7}\,\mathrm{Tg\,yr^{-1}}$, a

value that is is $6.3\,\mathrm{Tg\,yr^{-1}}$ larger than reflected in GFED. Our estimates are consistent with GFED at most other times. Wetland emissions are highest in 2015, which corresponds to a strong El Niño year. The *a priori* model emissions do not capture the increase in 2015 but do simulate a decrease from 2016. This feature is discussed further in Section 3.2.

The performance of the inversion is demonstrated through a comparison of modelled mole fractions derived from the posterior emissions and boundary conditions with the measurements used in inversion. We show this fit for both GOSAT and RPB

in Appendix Fig. A4 and we find both data sets to be represented well by the inversion.

### 3.2    Sub-national emissions

In addition to the Brazilian totals presented above, we aggregated our posterior emissions for the major regions of Brazil: the Amazon basin, the Pantanal and the remainder of the country (Figs. 5 and 6 for wetland and anthropogenic sectors, respectively). The Amazon basin was defined using the TRANSCOM definition for Tropical South America (Saunois et al., 2016)

and the Pantanal region was defined using the TRIP River Routing Model output (Oki et al., 1999). These regions were further

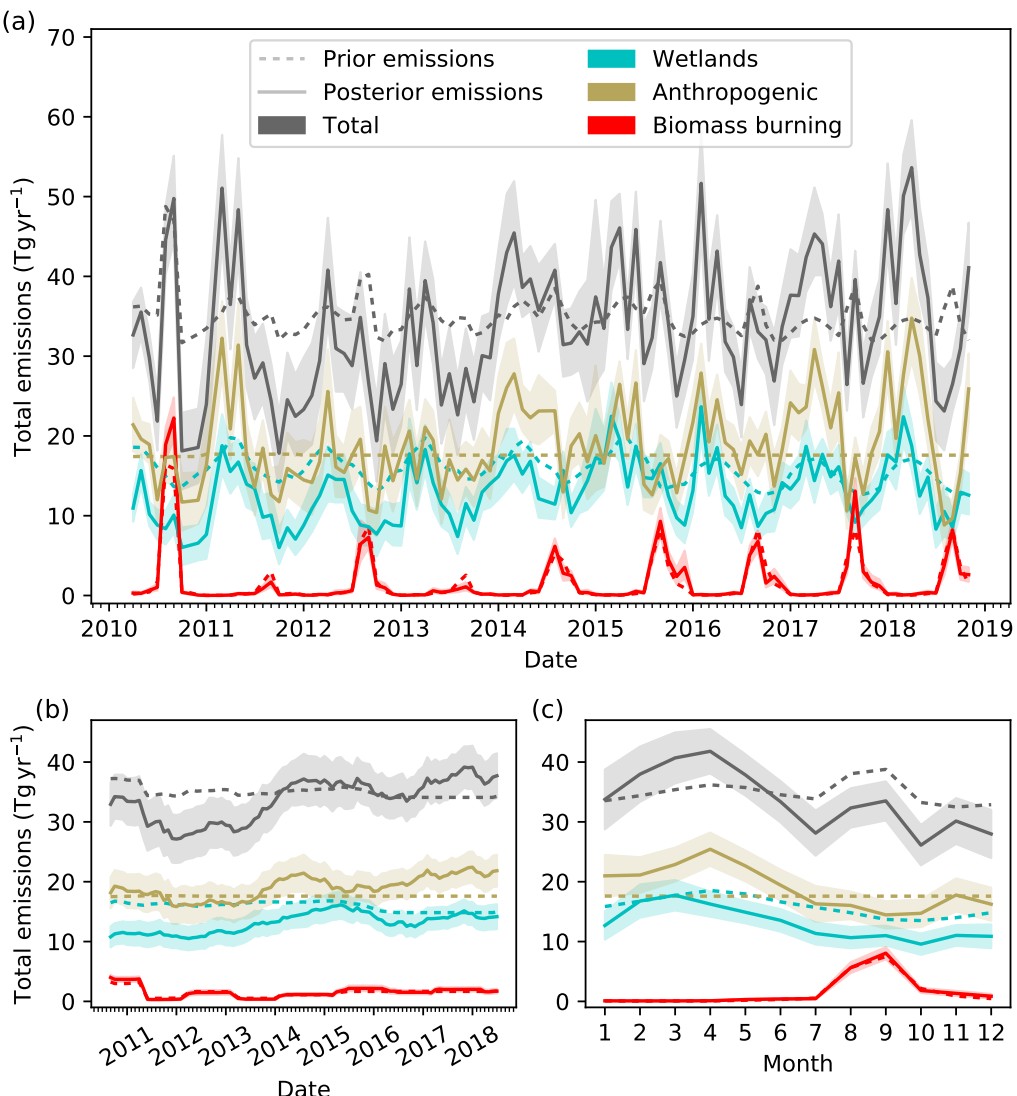

**Figure 4.** Brazil's $CH_4$ emissions derived from GOSAT and RPB measurements between 2010 and 2018. Total emissions (grey) are split into the three major sectors: wetlands (cyan), anthropogenic (yellow) and biomass burning (red). Prior and posterior emissions are dashed and solid lines, respectively. Shading indicates the 95% confidence interval. (a) Monthly emissions, (b) monthly emissions smoothed with a 12-month rolling mean and (c) monthly means across the 2010 to 2018 period. Errors in mean values assume a 50% correlation between individual months.

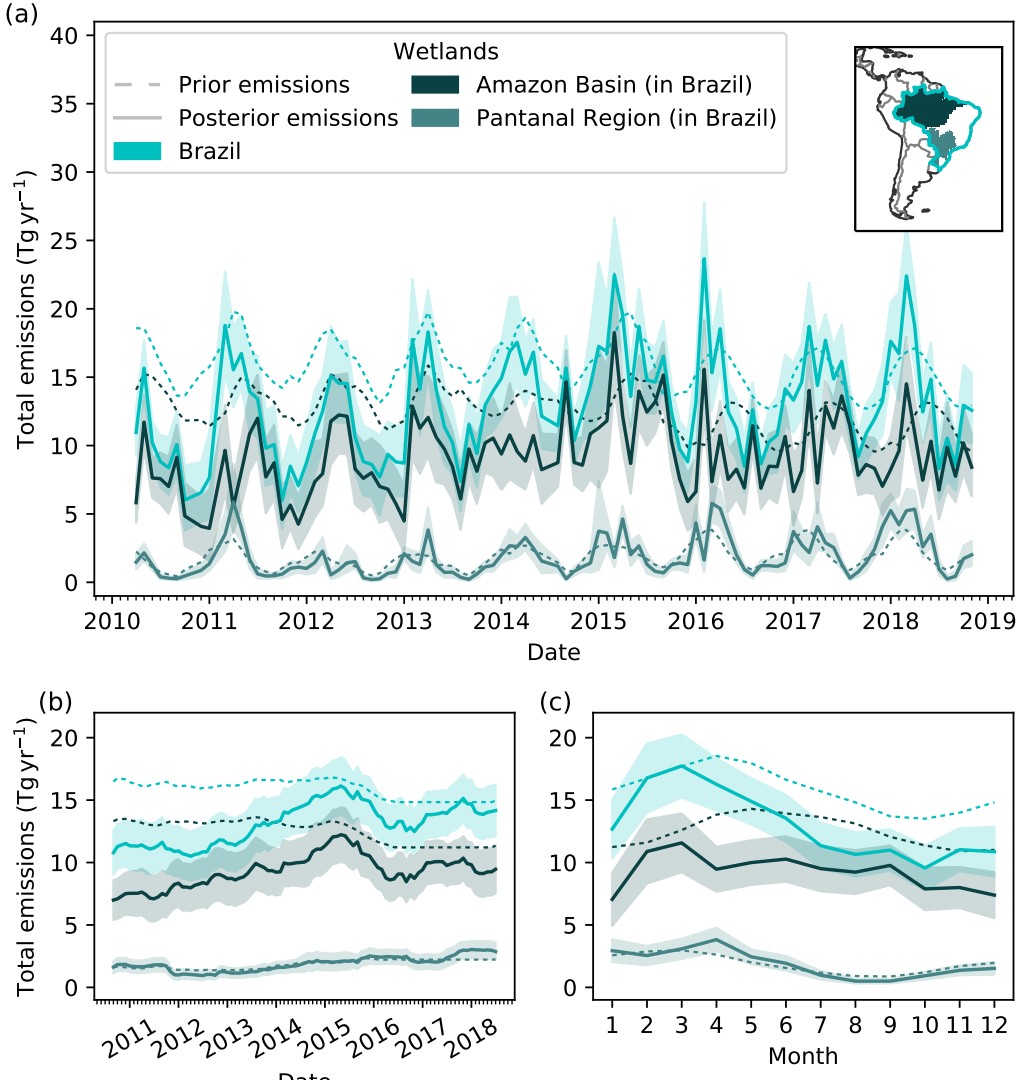

**Figure 5.** Brazil's wetland emissions aggregated over Amazon and Pantanal regions. (a) Monthly emissions with an inset map showing the masks used to delineate between the Amazon and Pantanal regions, (b) monthly emissions smoothed with a 12-month rolling mean and (c) seasonal means across the 2010 to 2018 period. Errors for mean values assume a 50% correlation between individual months.

masked to only include the area within Brazil using the public domain Natural Earth database (https://www.naturalearthdata.com/).

We aggregated wetland emissions (Fig. 5) into mean values for the 2010–2018 period, changes between 2011–2013 and 2014–2018, and means for each month. Mean wetland emissions from the Brazilian Amazon and Pantanal regions across the





2010–2018 period are $9.2 \pm 1.8\,\mathrm{Tg\,yr^{-1}}$ and $1.9 \pm 0.5\,\mathrm{Tg\,yr^{-1}}$, respectively. Wetland emissions in the Amazon and Pantanal comprise 65% and 26% of total emissions, respectively, with emissions from the Pantanal being dominated by the anthropogenic sector. While emissions from the Pantanal are not significantly different from the *a priori* emissions, these results are found to be robust in our sensitivity studies as discussed in Section 3.4. There is only a small change in wetland emissions over the two regions between 2011–2013 and 2014–2018. Differences are $1.5 \pm 2.6\,\mathrm{Tg\,yr^{-1}}$ and $1.0 \pm 0.7\,\mathrm{Tg\,yr^{-1}}$, for the Amazon

and Pantanal, respectively. Only the small Pantanal change is significant within the 95% confidence interval. We also find that there is an offset in peak emissions between the Amazon and the Pantanal regions. Amazon wetland emissions peak around February-March whereas the Pantanal peaks in April. The seasonality for the Amazon is earlier than reflected in the *a priori* emissions.

Because wetland emissions from the Pantanal exhibit a similar seasonal pattern to the seasonality in anthropogenic emissions
across Brazil, we analysed the regions that are responsible for driving the anthropogenic seasonal cycle. Figure 6 shows the anthropogenic emissions aggregated over the Amazon and Pantanal regions and over the remaining Brazilian territory. We find that the seasonal cycle is dominated by the emissions outside of the Amazon and the Pantanal. Therefore, while Pantanal wetland and anthropogenic emissions have the same seasonal pattern, this is not due to a mis-attribution between sectors based on the current configuration of the wetland and anthropogenic prior emissions. However, it is important to note the difficulty
of wetland models in capturing the full seasonal cycle in the Pantanal due to overbank inundation, so there could be some uncertainty in the fractional partitioning due to uncertainty in the wetland models used in the main results and in the sensitivity studies (Parker et al., 2018).

Wetland emissions are $3.9 \pm 3.0\,\mathrm{Tg\,yr^{-1}}$ larger in the 2015 wet season relative to 2011–2014, a feature that is not present in the *a priori* emissions. We show in Fig. 7 that this increase is driven from the Amazon and not by Pantanal wetlands.
We investigated changes in some of the major environmental influences to understand what could drive this pattern. Figure 7 shows our derived emission maps, changes in surface temperature from the WFDEI meteorological dataset (https://rda.ucar.edu/datasets/ds314.2/) and changes in SWAMPS inundation for the wet season, defined as February–April (FMA). We show differences between 2015 and 2011–2014 and between 2016 and 2015 for the Amazon and the Pantanal regions.

We find that the increase in 2015 originates mainly from the Western Amazon with a rise of $3.7 \pm 2.7\,\mathrm{Tg\,yr^{-1}}$ (defined as the
Brazilian Amazon area west of $-55°$E). This coincides with increased surface temperatures from this region. Wetland extent did not significantly change in the Western Amazon between 2015 and preceding years. Emissions then decrease after mid-2015 to levels that are sustained from 2016–2018. We find that this decrease is correlated with both lower soil temperatures and decreased inundation. The *a priori* emissions may be simulating the decrease after 2016 because the *a priori* emissions are constrained to the observational inundation fields. However, these results suggest that there may be uncertainties in the wetland
model temperature sensitivity.

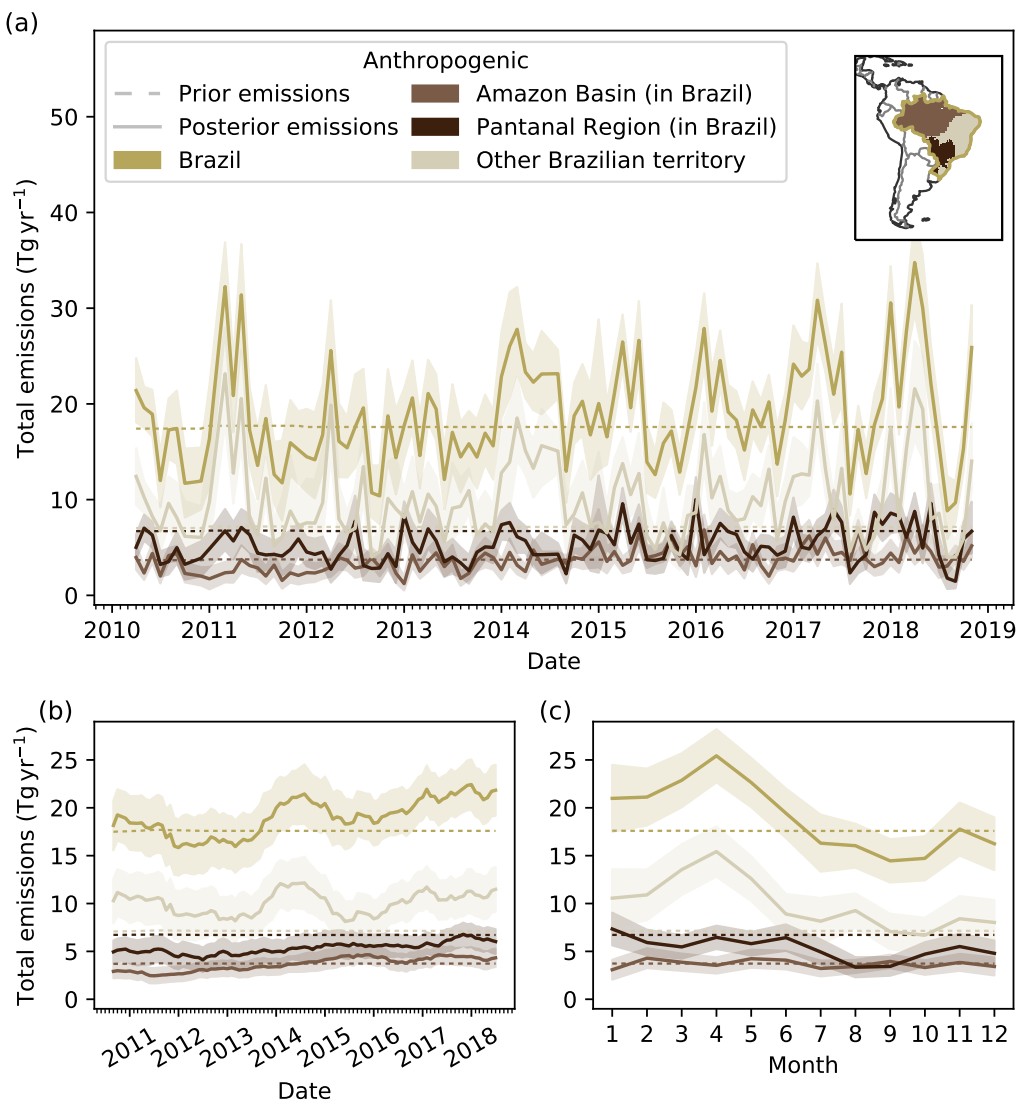

**Figure 6.** Brazil's anthropogenic emissions aggregated over the Amazon basin, Pantanal region and the rest of Brazilian territory. (a) Monthly emissions with an inset map showing the masks used to delineate different regions, (b) monthly emissions smoothed with a 12-month rolling mean and (c) seasonal means across the 2010 to 2018 period. Errors for mean values assume a 50% correlation between individual months.



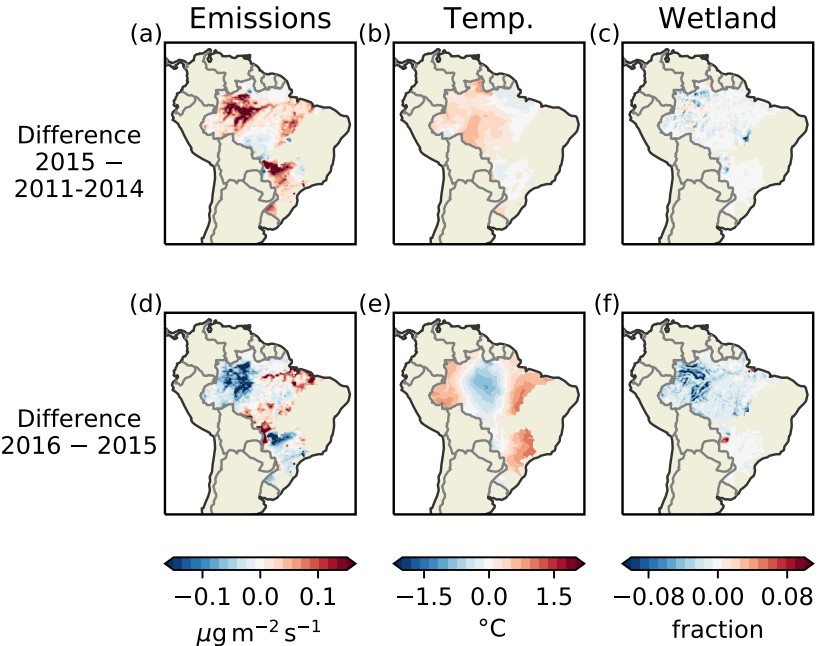

**Figure 7.** Differences during the wet season (FMA). (a-c) 2015 minus 2011–2014 average, and (d-e) 2016 minus 2015. (a,d) CH$_4$ emissions, (b,e) surface temperature from the WFDEI meteorological dataset, and (c,f) SWAMPS wetland fraction. All regions outside of the Brazilian Amazon and Pantanal have been masked for clarity.

### 3.3 Validation against ground-based data

We used independent data from ATTO to assess the robustness of our inversion results and to understand what factors are important for the inversion setup. The results of these tests can be seen in Fig. 8. Other datasets besides ATTO exist, such as aircraft data from the Amazon (Wilson et al., 2016; Pangala et al., 2017), but were not available for use.

An inversion using only GOSAT data produced a mean difference between modelled ATTO data and measurements of $42.5\,\mathrm{nmol\,mol^{-1}}$ (Figures 8a and 8e). This difference can largely be attributed to modelled boundary conditions that are consistently elevated throughout the year above the lowest ATTO data. Introduction of the surface baseline station of RPB (Figures 8b and 8f) improved the boundary condition estimation, with the modelled boundary conditions now consistent with ATTO data in most months and lower than ATTO data in months with significant regional emissions (i.e. times when ATTO may

not be representative of boundary conditions). Despite consistency with boundary conditions, this setup produced the highest mean difference with ATTO, $67.4\,\mathrm{nmol\,mol^{-1}}$, due to large regional emissions being estimated. The third case, the setup of our main results, which allowed for an offset between the GOSAT and RPB measurements to be estimated in the inversion, resulted in the best fit to ATTO (Figures 8c and 8g). The model achieved consistent boundary conditions and the smallest mean difference with ATTO ($18.9\,\mathrm{nmol\,mol^{-1}}$). In our inversions from 2010–2018, we estimate a mean offset parameter between




GOSAT and RPB data of $22 \pm 8\,\mathrm{nmol\,mol^{-1}}$. The numbers presented for the offsets are a combination of any bias between the data themselves, but also in the model's interpretation of these data sets. The model simulates the three-dimensional atmospheric fields necessary to combine these two datasets together. However, the interpretation of these tests show that near-surface data that help to constrain boundary conditions are required because GOSAT data alone does not have enough resolving power to partition boundary conditions and emissions. An offset parameter should then be included to account for a

combination of any differences between in-situ data and satellite data and any offsets due to the atmospheric model.

When our posterior emissions estimates were scaled to match previous results derived by Wilson et al. 2016 (but keeping the posterior boundary conditions fixed from our main results), a larger offset from ATTO of $45.4\,\mathrm{nmol\,mol^{-1}}$ (Figures 8d and 8h) again resulted. This test indicates that larger emissions from the Amazon are inconsistent with ATTO and its representation by the NAME model.

To assess the possibility of large systematic uncertainties in NAME, we show a comparison of the validation at ATTO generated using NAME (as in Figure 8c) with those generated using FLEXPART. This comparison is shown in Appendix Fig. A5 and shows that the posterior emissions and boundary conditions derived here are consistent with ATTO across both models. These results provide additional confidence in the magnitude of emissions that we derive.

### 3.4   Sensitivity studies

#### 3.4.1   Sensitivity to *a priori* emissions

Sensitivity tests to the effect of different wetland distributions are shown in Fig. 9. Total emissions do not change significantly between these sensitivity tests, despite the large seasonal cycle in the *a priori* WetCHARTS emissions that is not reflected in the other wetland distributions. This suggests that the inversion is well-constrained by the atmospheric data and is not significantly influenced by the prior. There are small differences in wetland and anthropogenic partitioning, but emissions are consistent

within uncertainties.

In addition to different *a priori* wetland distributions, sensitivity tests to perturb *a priori* emissions from each source sector are shown in Appendix Figures A6, A7 and A8. The main impact of perturbing *a priori* emissions from any source sector comes in the partitioning of total emissions into the sources, particularly between anthropogenic and wetland emissions. This is due to a small overlap between anthropogenic and wetland sources (Figures 2 and 3). However, the trade-off between these

two sectors is smaller than the initial perturbation to the prior and emissions are still consistent within confidence intervals, suggesting that the sectoral partitioning is robust. The largest sensitivity to the *a priori* emissions is shown when doubling *a priori* biomass burning emissions (Fig. A8) and the resulting posterior biomass burning estimate is not consistent within uncertainties to the unperturbed case. Overall, these tests show that our results and the associated sectoral partitioning, with the exception of some influence from the biomass burning prior, are robust to the *a priori* emissions used.



**Figure 8.** (a-d) Comparison of modelled and measured mole fractions at ATTO for different posterior emissions with (e-h) histograms of the differences. Posterior emissions derived in inversions using (a,e) GOSAT measurements only, (b,f) both GOSAT and RPB data but no offset parameter in the inversion, and (c,g) both GOSAT and RPB data and with an offset parameter between the two data sets in the inversion. (d,h) Posterior emission distribution of our main results but scaled so that total CH$_4$ emissions in the Brazilian Amazon was equivalent to those derived in Wilson et al. (2016).

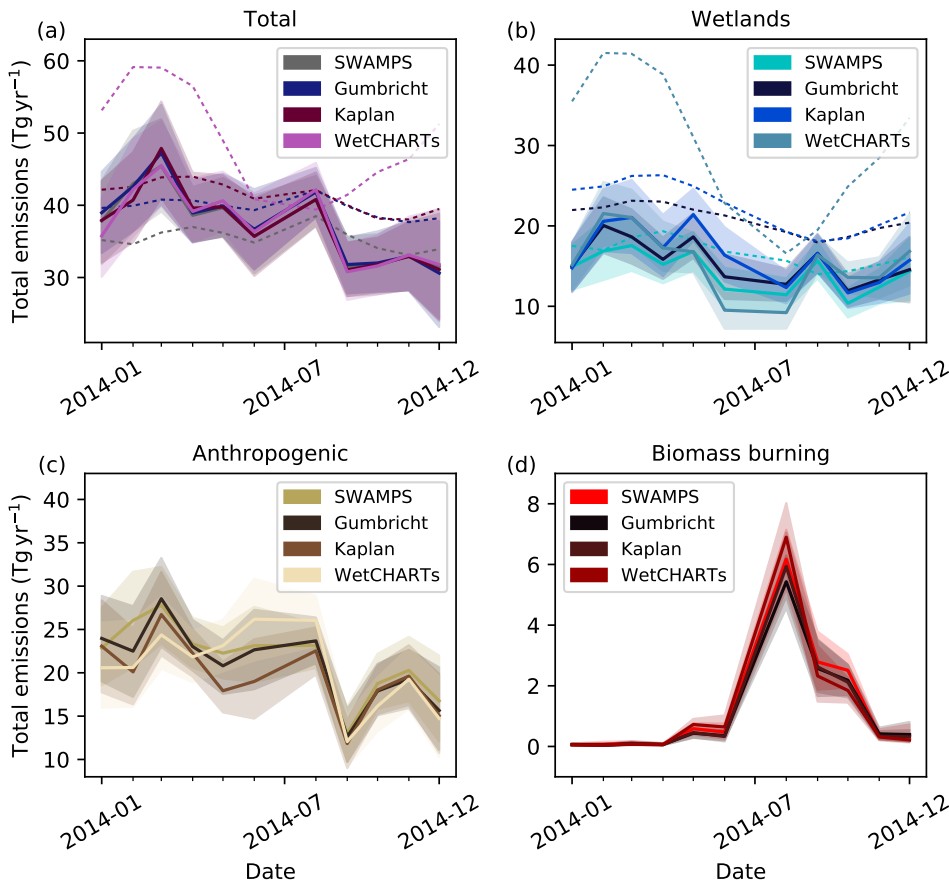

**Figure 9.** Brazil's CH$_4$ emissions in a sensitivity inversion using perturbed *a priori* wetland distributions. All other components were held at the original configuration. Total emissions for all of South America were scaled to $44\,\mathrm{Tg\,yr^{-1}}$ for all distributions apart from WetCHARTs, which used its derived emissions. (a) Total, (b) wetland, (c), anthropogenic, and (d) biomass burning emissions. Prior emissions are only shown in (a, b) as they only vary for these components in this test.

### 3.4.2 Influence of *a priori* boundary conditions

Results of using a different global model of *a priori* boundary conditions are shown in Appendix Fig. A9. Due to differences in the seasonal cycle when comparing the CAMS and MOZART boundary conditions in 2014, this analysis was run for the full 2010–2018 period to provide a longer comparison. While there is some month to month variability, the overall patterns are consistent between the two inversion setups, suggesting that the inversion is robust to the *a priori* boundary conditions. As demonstrated in Section 3.3, it is important to include data that can help the inversion constrain the boundary conditions, through, for example, surface measurements from remote background stations.





### 3.4.3 Influence of model XCO$_2$ on XCH$_4$

We generated ten variations of the XCH$_4$ dataset used in the inversion based on different model XCO$_2$. Figure 10 shows the emissions estimates that result when perturbing XCH$_4$ by random values of the model XCO$_2$ used to generate XCH$_4$ using the CO$_2$ proxy method. Because of some differences in 2014, this analysis was run for the full 2010–2018 period.


Across the ten variations, mean emissions over 2010–2018 range from 33.8–34.8 Tg yr$^{-1}$ in total, 19.0–19.4 Tg yr$^{-1}$ for anthropogenic, 13.0–13.4 Tg yr$^{-1}$ for wetlands and 1.7–1.8 Tg yr$^{-1}$ for biomass burning. Individual months can exhibit larger ranges in the ten variants, in some cases spanning $> 10$ Tg yr$^{-1}$. The differences based on model XCO$_2$ does not exhibit any particular seasonality. The change between the 2011–2013 and 2014–2018 periods across these ten inversions produces a range of 5.9–7.0 Tg yr$^{-1}$. Thus, the increase in emissions is robust to uncertainties in XCO$_2$.


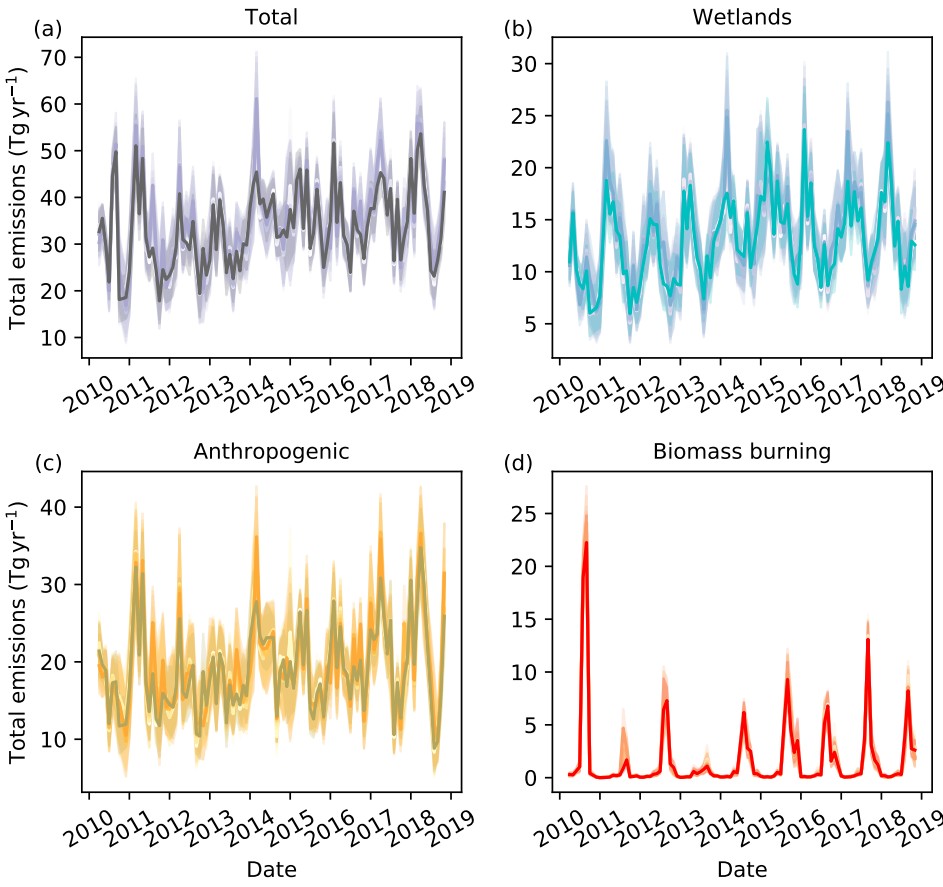

**Figure 10.** Brazil's CH$_4$ emissions in a sensitivity inversion using ten variants of GOSAT data generated with different model XCO$_2$. All other components were held at the original configuration. Each line represents a different inversion run. (a) Total, (b) wetland, (c), anthropogenic, and (d) biomass burning emissions. Prior emissions do not vary in this test.



## 4   Discussion

We find that Brazil's emissions increased during 2014–2018 over 2011–2013 levels by $6.9 \pm 5.3 \, \mathrm{Tg \, yr^{-1}}$ and this coincides with a large increase in global $CH_4$ mole fraction growth rate in 2014 (Nisbet et al., 2019). The increase in Brazil's emissions is primarily driven by anthropogenic and wetland sources. Brazil's anthropogenic emissions are dominated by agriculture and

mainly cattle, which is likely to be the main source for the inferred anthropogenic change. However, we did not have sufficient information with which to robustly separate total anthropogenic emissions into individual sub-sectors. Future work should couple measurements of $\delta^{13}$C-$CH_4$ from Brazil along with campaigns to sample representative isotopic source signatures (Ganesan et al., 2018) to better understand whether changes in these sources are consistent with isotopic constraints.

The increased wetland emissions that we derive in the wet season of 2015 primarily originates from the Western Amazon.

Previous studies have found that changes in wetland $CH_4$ emissions exhibit complex dynamics during El Niño years. Zhang et al. (2018) found through model simulations that the 2015–2016 El Niño led to larger instantaneous growth in $CH_4$ emissions than previous El Niño periods. This study also showed that there was a large increase in the Western Amazon due to increased soil respiration from high soil temperatures, despite a decline in wetland extent due to drought. This pattern is consistent with the results that we have derived from atmospheric data rather than from model simulations. In contrast to Zhang et al. (2018),

which also found a 2015–2016 El Niño effect on Western Amazon emissions, we find that emissions increased during the 2015 wet season rather than the 2016 wet season. We instead show a decline in 2016 emissions, surface temperature and wetland extent compared to 2015 levels. This discrepancy in temporal response from Zhang et al. (2018), suggests that the dynamics of the wetland response to climatic perturbations may require further investigation.

Our results show that emissions can be derived for a country of the size of Brazil from satellite data coupled with high-

resolution atmospheric transport modelling but careful consideration needs to be paid to the setup of the inversion. Regional inversions use atmospheric data to estimate boundary conditions and regional emissions. Due to the lower signal-to-noise of GOSAT data (which are sensitive to surface emissions that are mixed through the entire atmospheric column) compared to ground-based data (although the reduced surface sensitivity and precision of satellite data needs to be weighed against the greater geographical coverage), we find that additional surface data is required to better constrain the boundary conditions.

However, we find that when combining satellite data with calibrated surface data in an inversion it is critical to incorporate an offset parameter between the two datasets in the inversion. GOSAT data have been previously corrected by $7.7 \, \mathrm{nmol \, mol^{-1}}$ as a global average offset to independent ground based measurements from the Total Carbon Column Observing Network (TCCON) (Wunch et al., 2011), but large regional variations can exist from the global mean offset (Dils et al., 2014). This offset can be due to the data themselves as well as any offsets in the NAME model's simulation of the two datasets.

Janardanan et al. (2019) estimated Brazil's $CH_4$ emissions using a regional inversion method from 2011-2017 using GOSAT and surface data and find total emissions to be $56.2 \, \mathrm{Tg \, yr^{-1}}$ compared with $33.3 \pm 3.7 \, \mathrm{Tg \, yr^{-1}}$ derived in this study. The difference between our results can be attributed to the natural wetland emissions estimates for which Janardanan et al. (2019) derive $39.8 \pm 12.4 \, \mathrm{Tg \, yr^{-1}}$ compared to $13.1 \pm 1.9 \, \mathrm{Tg \, yr^{-1}}$ presented here. Anthropogenic estimates (excluding biomass burning) are similar at $16.5 \, \mathrm{Tg \, yr^{-1}}$ compared with our estimate of $18.8 \pm 2.6 \, \mathrm{Tg \, yr^{-1}}$. We propose that the main reason for the discrepancy





in our estimates stems from Janardanan et al. (2019) not including an offset parameter in their inversion. In the case where we similarly set up our inversion to not include an offset parameter (as shown by the ATTO comparison in Fig. 8b), we also derive larger total emissions of $51.7 \pm 3.5\,\mathrm{Tg\,yr^{-1}}$ for 2014. However, not allowing for this offset produces the poorest comparison to the independent ATTO measurements. However, it is important to note that our validation is based on only one site because of the availability of data.

Studies deriving Amazon Basin $CH_4$ emissions using aircraft data from within the Amazon are also higher than than our estimates at $49\,\mathrm{Tg}$ ($\mathrm{yr^{-1}}$, Miller et al. 2007; Wilson et al. 2016). However, these higher estimates, as shown in Fig. 8d, when simulated with NAME, are inconsistent when compared with $CH_4$ mole fractions measured at the ATTO tower. The wetland results presented here are most consistent with the lower bound estimates from Saunois et al. (2016) which range from 23.4–63.7 within Tropical South America. As discussed in Section 3.4.1, neither varying the magnitude of the prior input for

wetlands nor the wetland extent map used, significantly altered our posterior estimates.

We propose one reason for the difference from aircraft based estimates could be that the studies using aircraft data may not be able to constrain emissions over the whole of the Amazon Basin and furthermore, at the country-scale, though our comparison at present has only been validated by one in situ measurement station. Future work should perform a detailed comparison between aircraft-derived estimates and those derived from satellites, investigating the inversion setup and the

degree of constraint by the datasets. The main benefit of using satellite data is in its widespread coverage, which allows for country-scale emissions to be derived (albeit, with inclusion of calibrated near-surface data in the inversion).

Overall, we derive lower emissions than previous studies. We show the validation of our results at ATTO using two models, NAME and FLEXPART. The consistency between the two models in simulating the magnitude of mole fractions at ATTO provides some confidence in the lower emissions we derive over previous studies. In future, performing a full set of inversion

results using a large range of models with different physical parameterisations could help to quantify the magnitude of any systematic uncertainties.

## 5 Conclusions

We estimated Brazil's $CH_4$ emissions from 2010–2018 using a combination of GOSAT satellite data and surface data from Ragged Point, Barbados. Due to the spatial and temporal separation in the three main sources of Brazil's emissions (anthro-

pogenic, wetland and biomass burning), we were able to derive emission estimates by sector.

We find mean emissions from 2010–2018 to be $33.6 \pm 3.6\,\mathrm{Tg\,yr^{-1}}$, corresponding to $19.0 \pm 2.6\,\mathrm{Tg\,yr^{-1}}$ from anthropogenic, $13.0 \pm 1.9\,\mathrm{Tg\,yr^{-1}}$ from wetland and $1.7 \pm 0.3\,\mathrm{Tg\,yr^{-1}}$ from biomass burning. We find a rise of $6.9 \pm 5.3\,\mathrm{Tg\,yr^{-1}}$ occurring between 2011–2013 and 2014–2018 periods. Both anthropogenic and wetland sources drive the increase in emissions over the period. This rise in emissions occurred during a period of accelerated global $CH_4$ growth, suggesting that Brazil's $CH_4$ sources

have a significant influence on changes in the atmosphere.



We find that wetland emissions from the Western Amazon increased by $3.7 \pm 2.7 \, \mathrm{Tg \, yr^{-1}}$ in the 2015 wet season, at the start of the 2015–2016 El Niño, and decreased subsequently from 2016. We show that the increase is likely to be driven by increased surface temperatures and thus, respiration rates, rather than through changes in inundation.

Our study demonstrates that satellite data, with its enhanced coverage compared to surface data, can be used to infer country-scale emissions. This is beneficial for independently comparing top-down estimates with national reports to the UNFCCC. However, we show that satellite data must be used in conjunction with calibrated surface data, which provide critical constraints on boundary conditions in regional inversions. It is also necessary to account for any offsets between datasets which can result from either biases between satellite data and surface data or from the atmospheric transport model used to simulate these data. Otherwise the resulting emission estimates may be biased. Our sensitivity studies show that our emissions estimates are insensitive to most inputs, but the largest differences are driven by uncertainties in the model $XCO_2$ used to derive $XCH_4$.

*Code and data availability.* University of Leicester GOSAT Proxy $XCH_4$ data can be accessed via the Copernicus Climate Data Store or by contacting Rob Parker. RPB data can be accessed from https://data.ess-dive.lbl.gov/view/doi:10.3334/CDIAC/ATG.DB1001 and by contacting Dickon Young. ATTO data can be accessed from https://www.attodata.org/ and by contacting Jošt Lavrič. The inversion code and NAME footprints used in this study can be accessed by contacting Rachel Tunnicliffe and Anita Ganesan.

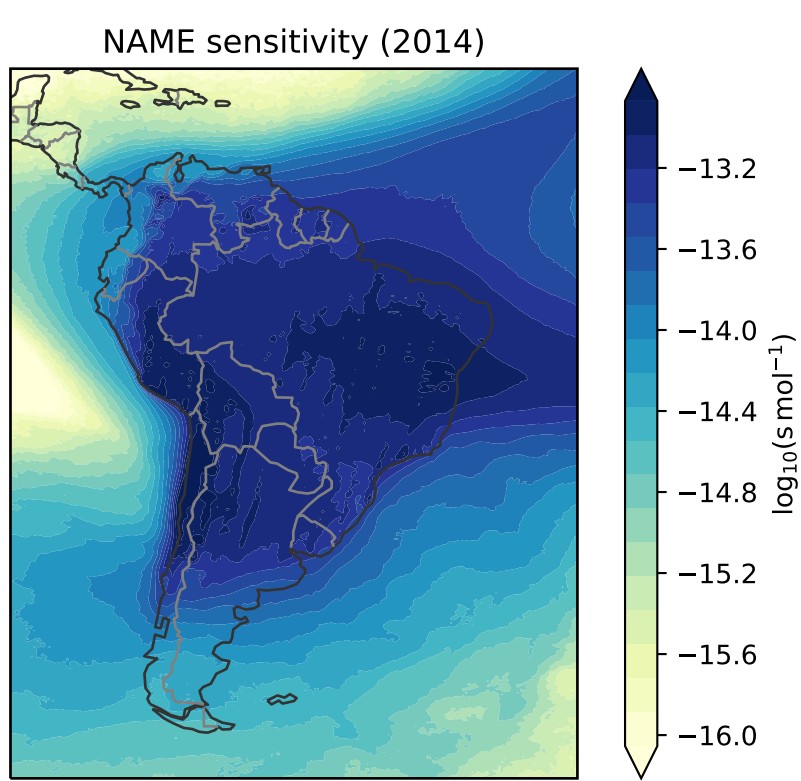

**Figure A1.** Annual mean NAME sensitivity map for GOSAT measurements in nadir mode within the area $-35.8$ to $7.3°$N and $-76.0$ to $-32.8°$E over Brazil, for 2014.

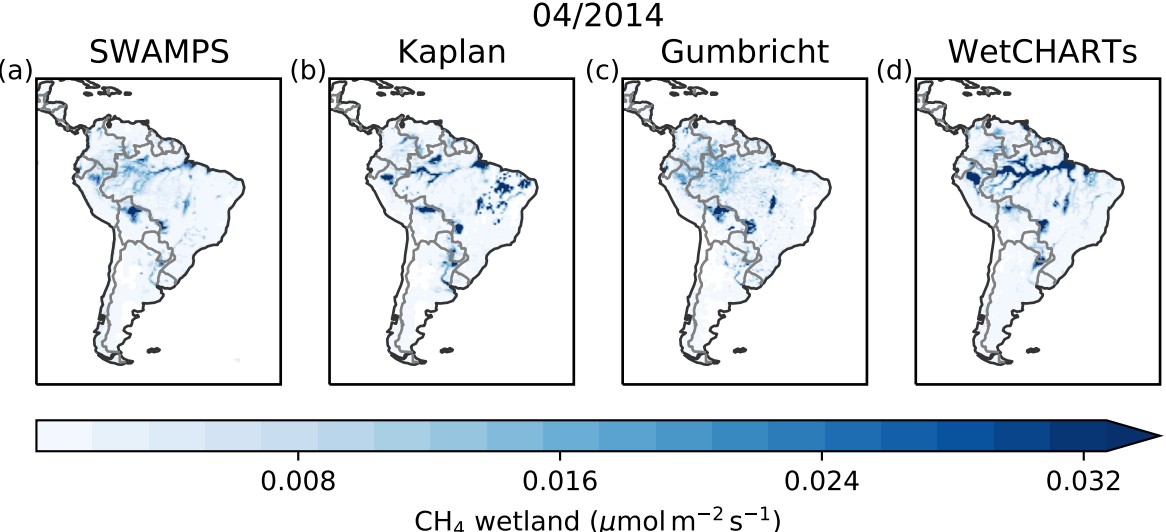

**Figure A2.** CH$_4$ emissions for each of the four *a priori* wetland emissions used in the wetland extent sensitivity study. Panels (a)-(c) SWAMPS, Kaplan and Gumbricht fractional maps are combined with the JULES emissions output. Details of these inversion setups are described in Table 2. This is shown for April, 2014 which is a wet season month with high emissions in the Amazon basin and the Pantanal region.



**Figure A3.** Emissions maps across 2011–2018 time period grouped into February-April (FMA), May-July (MJJ), August-October (ASO) and November-January (NDJ). Panels (a)-(d) CH₄ posterior emissions maps and (e)-(h) difference between the posterior and the *a priori* emissions input.

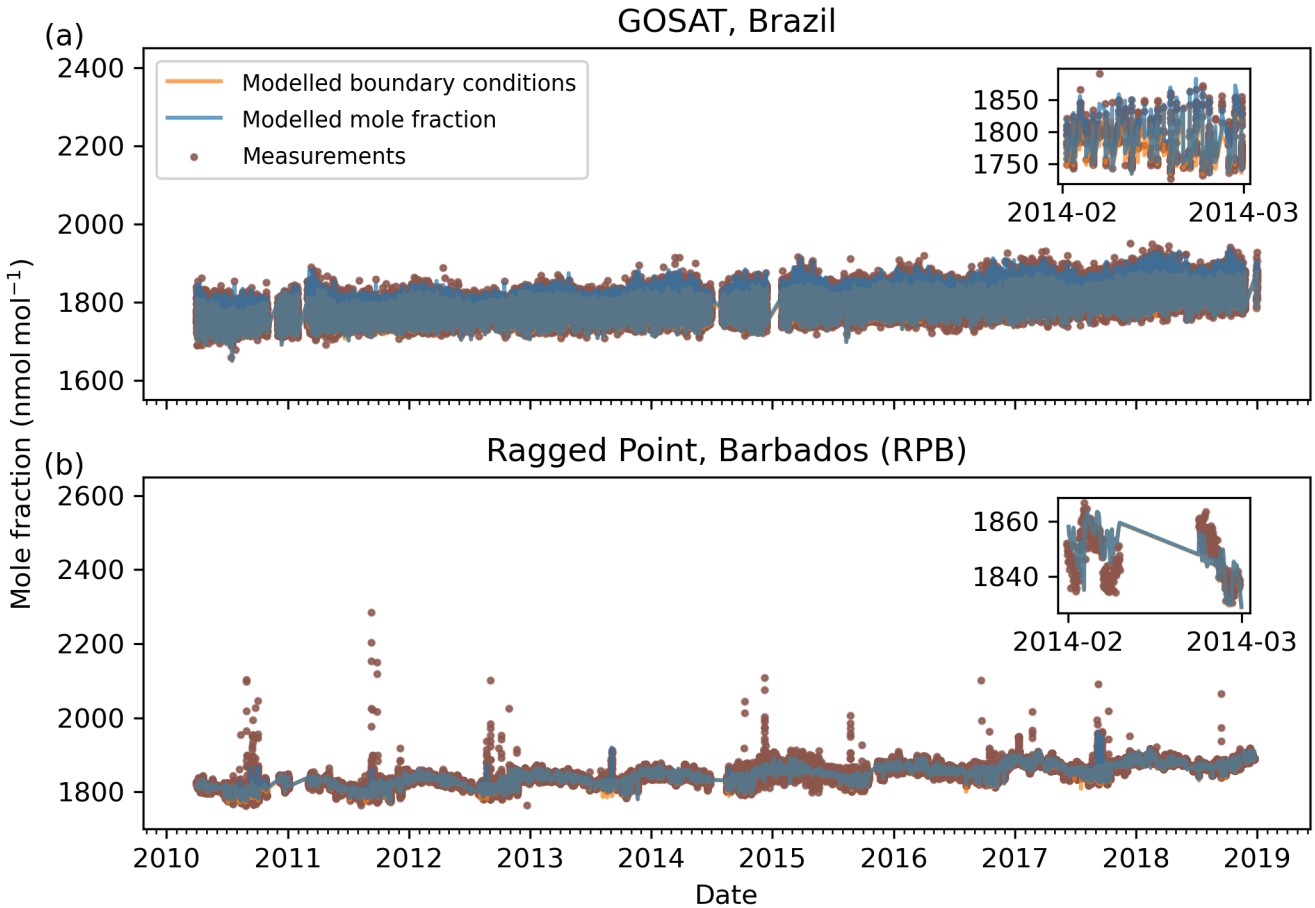

**Figure A4.** Modelled mole fractions derived from the posterior emissions estimate compared to measurements from (a) GOSAT and (b) RPB. The orange line shows posterior boundary conditions and the blue line shows the total modelled mole fraction. Measurements are displayed as red dots.

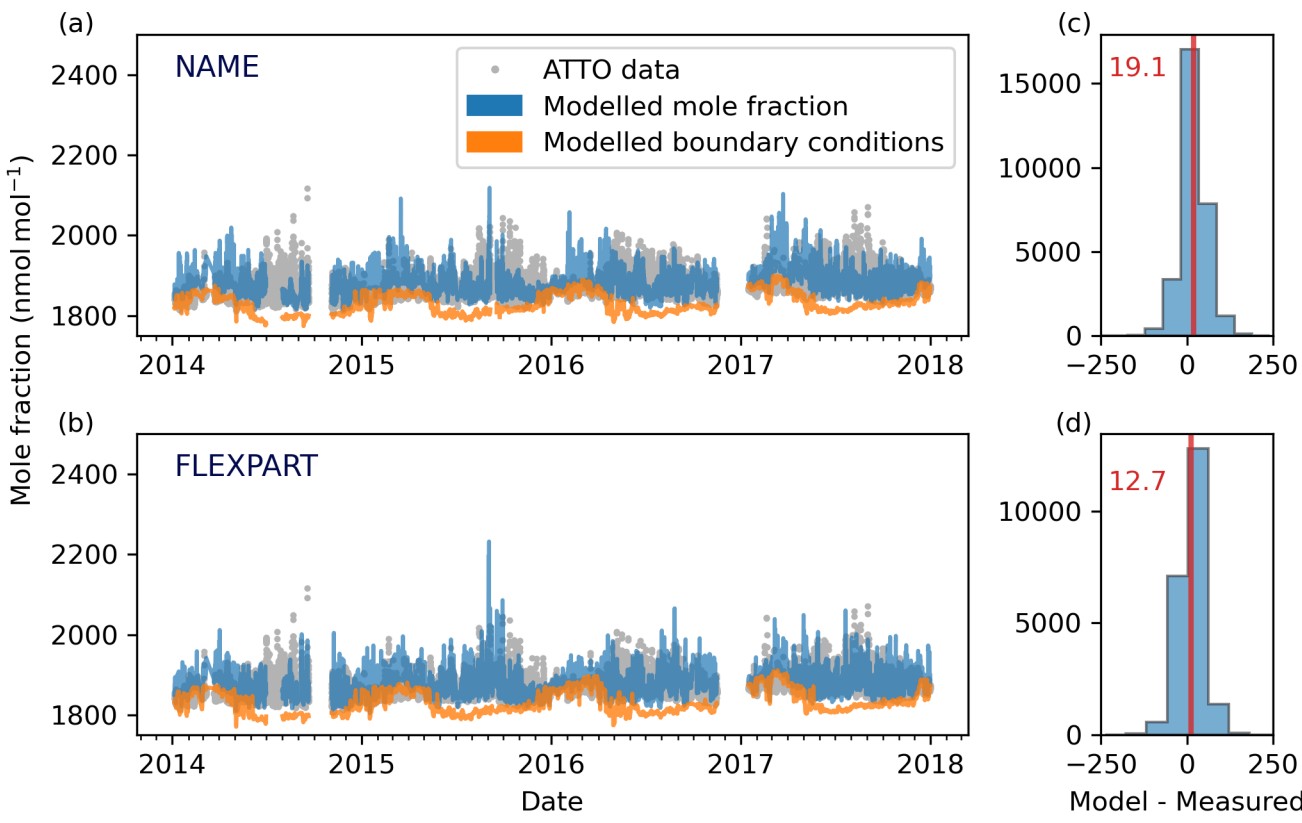

**Figure A5.** Comparison of modelled and measured mole fractions at ATTO with the same posterior emissions and boundary conditions convolved with sensitivity maps derived from two different models. (a) NAME model (main results) and (b) FLEXPART model where (c-d) shows histograms of the difference. Posterior emissions were derived from our inversion setup using both GOSAT and RPB data with an offset parameter between the two data sets allowed within the inversion.



**Figure A6.** Brazil's CH$_4$ emissions in a sensitivity inversion using a perturbed *a priori* wetlands emissions magnitude. All other components were held at the original configuration. *A priori* wetland emissions for all of South America were scaled to 32 (low), 44 (mean) or 50 (high) Tg yr$^{-1}$ (as defined for Tropical South America in Saunois et al. 2016). (a) Total, (b) wetland, (c), anthropogenic, and (d) biomass burning emissions. Prior emissions are only shown in (a, b) as they only vary for these components in this test.





**Figure A7.** Brazil's CH$_4$ emissions in a sensitivity inversion using perturbed *a priori* anthropogenic emissions. All other components were held at the original configuration. Anthropogenic emissions for all of South America were doubled from EDGAR. (a) Total, (b) wetland, (c), anthropogenic, and (d) biomass burning emissions. Prior emissions are only shown in (a, c) as they only vary for these components in this test.



**Figure A8.** Brazil's CH$_4$ emissions in a sensitivity inversion using perturbed *a priori* biomass burning emissions. All other components were held at the original configuration. Biomass burning emissions for all of South America were doubled from GFED. (a) Total, (b) wetland, (c), anthropogenic, and (d) biomass burning emissions. Prior emissions are only shown in (a, d) as they only vary for these components in this test.

**Figure A9.** Brazil's CH$_4$ emissions in a sensitivity inversion using perturbed *a priori* boundary conditions from the MOZART model. All other components were held at the original configuration. (a) Total, (b) wetland, (c), anthropogenic, and (d) biomass burning emissions. *A priori* emissions do not vary in this test.



*Author contributions.* R.T and A.G designed the methodology and wrote the manuscript. R.T performed the analysis. R.P and H.B provided the GOSAT data. N.G provided JULES $CH_4$ emission fields. B.P. and Z.Z. provided SWAMPS wetland extent maps. J.L and D.W. provided data from the Amazon Tall Tower Observatory. M.R advised on methodology. D.Y and S.O. provided data from Ragged Point, Barbados.

*Competing interests.* We declare no conflicts of interest.

*Acknowledgements.* This work and its contributors (R.T) were supported by the Newton Fund through the Met Office Climate Science for Service Partnership Brazil (CSSP Brazil) and the Natural Environment Research Council (NERC) Methane Observations and Yearly Assessments programme (MOYA, NE/N016548/1). A.G was funded by the NERC Independent Research Fellowship NE/L010992/1. R.J.P and H.B were funded via the UK National Centre for Earth Observation (NCEO grant number: nceo020005).

We thank the Japanese Aerospace Exploration Agency, National Institute for Environmental Studies, and the Ministry of Environment for
the GOSAT L1B data and their continuous support as part of the Joint Research Agreement. We thank Alistair Manning (Met Office) for useful and insightful discussions about this work.

The operation of the Ragged Point site was funded by the National Aeronautical and Space Administration (NASA, USA) (grants NAG5-12669, NNX07AE89G and NNX11AF17G to MIT; grants NAG5-4023, NNX07AE87G, NNX07AF09G, NNX11AF15G and NNX11AF16G to SIO) under the AGAGE programme and the National Oceanic and Atmospheric Administration (NOAA, USA) (contract RA-133R-15-CN-
0008 to the University of Bristol). ATTO data were supported by the Max Planck Society (MPG), the German Federal Ministry of Education and Research (contracts 01LB1001A and 01LK1602A) and the Brazilian Ministério da Ciência, Tecnologia e Inovação (MCTI/FINEP contract 01.11.01248.00) as well as the Amazon State University (UEA), FAPEAM, LBA/INPA and SDS/CEUC/RDS-Uatumã. We acknowledge the Swiss Federal Office for Meteorology and Climatology (MeteoSwiss) for providing access to ECMWF ERA-Interim reanalysis products for use with the FLEXPART model. This work was carried out using the computational facilities of the Advanced Computing Research
Centre, University of Bristol - http://www.bristol.ac.uk/acrc/. GOSAT retrievals used the ALICE High Performance Computing Facility at the University of Leicester.





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
