# Peer review of "Quantifying sources of Brazil's CH4 emissions between 2010 and 2018 from satellite data"

_Atmospheric Chemistry and Physics, 2020_

## Short Comment (SC1) · 29 Jun 2020

Please correct the affiliation under nr. 8 - it should read "Max Planck Institute for Biogeochemistry, Jena, Germany".

Thanks

---

## Short Comment (SC2) · 30 Jun 2020

Thanks for pointing out the mistake in the affiliation - I'll make sure that gets corrected.
* * *

---

## Referee Comment (RC2) · Anonymous Referee #2 · 15 Jul 2020

General Comments:

The authors present a detailed top-down quantification of methane emissions from Brazil. They use GOSAT satellite observations to estimate sectoral and regional emissions at a monthly temporal resolution. The analyses are performed in a thorough manner and include multiple sensitivities tests. The inversion estimates larger emissions from Brazil during 2014-2018 than during 2011-2013, which could have contributed to the accelerated global methane growth rate from 2014. The robustness of emission estimates derived here gives confidence in the capability of satellite observations—-which suffer from coverage issues over the tropics due to clouds—-to provide good emission quantifications from tropical regions. This study provides a demonstration of how the rapidly expanding satellite observation dataset can be used to constrain country scale

emissions, which can aid in emission reporting and monitoring. The manuscript is well written, with clearly presented results, and it is suitable for publication after some minor issues are addressed.

Minor comments:

Line 124: A uniform distribution ranging from 0.2 to 200 nmol mol-1 is used to define the model-measurement PDF. Does this mean that the inversion does not allow the model mixing ratios to be lower than measurements? Why not use a PDF centred around zero?

Line 126: Are there PDFs of the two offsets? I assume that they would be needed for the inversion to decide the relative weights of the emission vs offset adjustments. Or are the offsets evaluated before the emissions in a separate step? Please clarify.

Line 190: Impact of model-CO2 on the proxy-XCH4 dataset is crucial for regions like Brazil as they can have strong CO2 emission interannual variabilities, which would impact proxy-XCH4. The CO2 models assimilating only surface observations might not capture such variabilities well due to lack of surface observation in the region. One way to address this would be to compare the full-physics XCH4 data with the proxy data for differences in interannual variabilities.

Line 216 to 220: The September 2010 biomass burning emissions difference of between GFED and the inversion is not a good example of "Our analysis shows that individual years exhibit features that are not present in the bottom-up estimates." as both estimates show 2010 has the highest biomass burning emissions and emission peaks in September.

Line 205: The authors write "The inversion results show that the Biomass burning emission rose by 1 ± 0.4 Tg/yr between 2011-2013 to 2014-2018". Can this be checked in the GFED data of more recent years?

Technical Corrections:

line 11: The sentence is difficult to understand. Writing it as an inline list will make it easier to read.

Line 220: remove the double "is"

Line 345: "modelling but" => "modelling, but"

---

## Short Comment (SC3) · 15 Jul 2020

The publication by Janardanan et al., (2020) was referred to as a regional inversion by the authors in the 'Introduction' and 'Discussion' sections. Janardanan et al., (2020) presented a global inversion system with discussion on regional/country scale. Also, the paper was published in 2020, not 2019. I would like to bring these to the kind attention of the authors.

Thank you very much. Rajesh janardanan

---

## Author Comment (AC1) · 21 Aug 2020

Please also see attached PDF which contains the same comments with formatting.

We would like to thank the reviewer for their valuable comments and include responses below. Line numbers have been included where appropriate.

Response to Reviewer #1

The authors presented an inverse modeling study of Brazil's methane emissions using observations by GOSAT satellite. Their estimate of anthropogenic emissions matches with Brazil's national inventory, while estimated emissions from wetlands are smaller than those by several other studies. To check the validity of the results and to quantify the impact of uncertainties in the inputs, the authors implemented a number of sen-

sitivity studies. Although the study doesn't use the ground-based observations inside the target region in the inversion, the results are supported by sensitivity tests. Discussions point out a large uncertainty of wetland emissions and the spread of different estimates, which has to be investigated further in the future. Paper is well written and can be accepted after minor revisions reflecting the review comments.

Detailed comment.

1) Line 50-56, Authors try to show that there is a wide range of estimates. To make that point it's better to group together the estimates for same regions/categories. It is not clear how big is the difference between studies when the target area is different. Another study using aircraft observations by Beck at al. (2013) could also be mentioned.

The discussion around previous results has now been rearranged to focus on total emissions rather than mixing both total and wetland emissions to allow these numbers to be more easily contrasted while considering the different regions that encompass this work. Reference to Beck et al. 2013, which combined modelling with aircraft measurements over the Amazon, has also been added when discussing these results.

The text has now been updated as follows: [Lines 50-59] Current top-down estimates of $CH_4$ emissions from Brazil, the Amazon and tropical South America vary depending on the method, source of data and area considered. In the synthesis of Saunois et al. 2016, across the Tropical South America region, emission estimates derived using different datasets and top-down methods span the large range of 63 - 119 Tg yr-1 (23 - 69 Tg yr-1 from wetlands) for 2012. Across the Amazon basin, estimates of total emissions derived from aircraft measurements are between $\sim 16 - 72$ Tg yr-1 derived for May, 2009 (Beck et al. 2013) and $31 - 43$ Tg yr-1 for 2010 - 2013 (Wilson et al. 2016; Pangala et al. 2017). A recent study that performed a regional analysis using satellite data by Janardanan et al. 2020 found Brazil's emissions alone, on average, to be 56.2 Tg yr-1 (39.8 +/- 12.4 Tg yr-1 from wetlands) across 2011-2017. In addition, many previous studies have estimated emissions globally using satellite data (e.g.,

**[ACPD](ACPD)**
Bergamaschi et al. 2009; Feng et al. 2017). The wide range of estimates indicate that large uncertainties exist and these uncertainties are exacerbated when estimating emissions over smaller scales such as the Amazon basin or when quantifying individual sources.

2) Line 53-54 Janardanan et al (2019) used global inversion, not regional. The correct publication year is 2020, not 2019.

Thank you for spotting this error. All references to Janardanan et al. 2019 have been updated to Janardanan et al. 2020 [Lines 59, 360, 363, 367]

The text has now been updated as follows: [Line 362] Janardanan et al. 2020 estimated Brazil's CH4 emissions using a coupled global Eulerian–Lagrangian model from 2011 - 2017 using GOSAT and surface data and find total emissions to be 56.2 Tg yr-1 compared with 33.3 +/- 3.7 Tg yr-1 derived in this study.

3) Line 270-275 The discussion implies that there is a bias in boundary conditions (taken from global models). Is there any bias between those global models and data at RPB?

The offset parameter used in the inversion does not represent a bias between Ragged Point and the global model used to produce the boundary conditions but represents any biases between the Ragged Point and GOSAT data used in the inversion as well as any biases in how the atmospheric transport model is able to simulate the two datasets (one surface, one column).

In fact, the CAMS inversion product assimilated surface sites including Ragged Point, and thus by design, there is consistency between Ragged Point data and the CAMS global mole fraction product. However, we include offsets to the global mole fraction field in our regional inversion to account for any variations that might occur outside of the assimilated data points in CAMS.

4) Line 290-295 Figures 8 and A5 show the observation and model time series, while it

is difficult to understand the sign of mean mismatch between observed and simulated concentrations. It would be useful to add monthly mean data to make differences easier to see.

For clarity, figures 8 and A5 have been updated to include the monthly means of both the modelled and measured mole fractions. In addition, the histogram already included gives a summary of the size and direction of the offset between the modelled and measured data. These two updated figures have been attached.

5) Line 355 The reasons for different models to give diverting results could be low number of GOSAT observations in wet season over the western Amazon basin. The full physics algorithm retrievals are likely to produce less data than proxy retrievals in partly cloudy conditions.

We have added a statement regarding estimates derived from different retrieval algorithms, which can impact both the retrieved mole fractions as well as the different numbers of data points.

The text has now been updated as follows: [Line 364-369] The difference between our results can be attributed to the natural wetland emissions estimates for which Janardanan et al. 2020 derive 39.8+/-12.4 Tg yr-1 compared to 13.1 +/- 1.9 Tg yr-1 presented here. Anthropogenic estimates (excluding biomass burning) are similar at 16.5 Tg yr-1 compared with our estimate of 18.8 +/- 2.6 Tg yr-1. One factor in this difference could be the differing GOSAT retrieval products used which were derived using different algorithms (CO2 proxy vs full physics retrievals). Another reason for the discrepancy could stem from Janardanan et al. 2020 not allowing for an offset parameter between the surface/aircraft and satellite data within their inversion.

6) Line 359 Most global inversions add a latitude-dependent offset to XCH4 in a way proposed by Bergamaschi et al (2009). So, it is better to note that offset is added differently here.

The GOSAT product we are using has already had a global bias correction applied (7.7 nmol mol-1) based on the global average difference with TCCON. The offset that we are describing in reference to the Janardanan et al. 2020 is due to any additional regional differences between the datasets as well as in their representation by the NAME model. The text has been updated to reflect the inclusion of the latitudinal correction in other inversions [Line 357] and text around the comparison with Janardanan et al. 2020 has been modified to clarify this [Line 367].

The text has now been updated as follows: Line [356-361] The GOSAT product used here has been previously corrected by 7.7 nmol mol-1 as a global average offset to independent ground based measurements from the Total Carbon Column Observing Network (TCCON) (Wunch et al. 2011). However, large regional variations can still exist (Dils et al. 2014). The offset estimated in an inversion is due to biases between the different datasets as well as their representation by the atmospheric transport model. Other inversion studies have imposed latitude-dependent bias corrections to the GOSAT data (e.g. Bergamaschi et al. 2009, Turner et al. 2015). Line [366-368] – see correction (5) for this updated text

7) Line 365-370 High tropical wetland emissions are needed in global models to fit the observations. When there are observations downwind of Amazon basin such as aircraft data used by Wilson et al. (2016), discarding those estimates as improbable needs to be done with some caution.

It is not our intention to suggest disregarding the estimates produced using aircraft data but simply to consider how best to reconcile the difference in our results with past results. When describing these results as inconsistent [Line 292-293] this is done in relation to the ATTO tall tower data and its representation by the NAME/FLEXPART models.

The text has now been updated as follows: Line [381-383] Future work should perform a detailed comparison between aircraft-derived estimates and those derived from

satellites, investigating the inversion setup and the degree of constraint by the datasets to understand the reasons for this discrepancy. Line [374-375]: However, these higher estimates, as shown in Fig. 8d, when simulated with NAME, are less consistent when compared with CH4 mole fractions measured at the ATTO tower.

References • Beck, V., Gerbig, C., Koch, T., Bela, M. M., Longo, K. M., Freitas, S. R., Ka-plan, J. O., Prigent, C., Bergamaschi, P., and Heimann, M.: WRF-Chem simula-tions in the Amazon region during wet and dry season transitions: evaluation of methane models and wetland inundation maps, Atmos. Chem. Phys., 13, 7961–7982,https://doi.org/10.5194/acp-13-7961-2013, 2013. Reference to Beck et al. 2013 has been added (see point 1)

Please also note the supplement to this comment:
https://acp.copernicus.org/preprints/acp-2020-438/acp-2020-438-AC1-supplement.pdf
* * *
[Figure]

[Figure]

Fig. 1.

[Figure]

Fig. 2.

---

## Author Comment (AC2) · 21 Aug 2020

Please also see attached PDF which includes the same comments with formatting.

We would like to thank the reviewer for their valuable comments and include responses below. Line numbers have been included where appropriate.

Reviewer #2 The authors present a detailed top-down quantification of methane emissions from Brazil. They use GOSAT satellite observations to estimate sectoral and regional emissions at a monthly temporal resolution. The analyses are performed in a thorough manner and include multiple sensitivities tests. The inversion estimates larger emissions from Brazil during 2014-2018 than during 2011-2013, which could have contributed to the accelerated global methane growth rate from 2014. The robustness of

emission estimates derived here gives confidence in the capability of satellite observations - which suffer from coverage issues over the tropics due to clouds - to provide good emission quantifications from tropical regions. This study provides a demonstration of how the rapidly expanding satellite observation dataset can be used to constrain country emissions, which can aid in emission reporting and monitoring. The manuscript is well written, with clearly presented results, and it is suitable for publication after some minor issues are addressed.

Minor comments:

1) Line 124: A uniform distribution ranging from 0.2 to 200 nmol mol-1 is used to define the model-measurement PDF. Does this mean that the inversion does not allow the model mixing ratios to be lower than measurements? Why not use a PDF centred around zero?

We apologize for a poorly phrased sentence. To clarify, the model-measurement uncertainty is represented with a Gaussian PDF centred on zero with a standard deviation that is governed by a hyper-parameter with a range of values allowed from 0.2-200 nmol mol-1.

The text has now been updated as follows:

Line [124-126] The model-measurement uncertainty was governed by a Gaussian distribution centred on zero and with a standard deviation that was a hyper-parameter in the inversion. The standard deviation hyper-parameter was described by a uniform distribution with a range of 0.2 to 200 nmol mol-1.

2) Line 126: Are there PDFs of the two offsets? I assume that they would be needed for the inversion to decide the relative weights of the emission vs offset adjustments. Or are the offsets evaluated before the emissions in a separate step? Please clarify.

The offsets are both represented in the inversion with Normal PDFs. The text describing this has been expanded to include these details (with associated parameters).

The text has now been updated as follows:

Line [128-133] In addition, an offset parameter was included to account for any differences between the satellite and the calibrated ground-based measurements and their representation by models. A normal PDF was defined for both of these types of offsets, centered on zero, and where the standard deviation of PDFs were governed by hyper-parameters. The standard deviations of the boundary condition offsets were allowed to vary up to up to 100 nmol mol-1 and up to 50.0 nmol mol-1 for the offset between surface and satellite data.

3) Line 190: Impact of model-CO2 on the proxy-XCH4 dataset is crucial for regions like Brazil as they can have strong CO2 emission interannual variabilities, which would impact proxy-XCH4. The CO2 models assimilating only surface observations might not capture such variabilities well due to lack of surface observation in the region. One way to address this would be to compare the full-physics XCH4 data with the proxy data for differences in interannual variabilities.

We agree that our sensitivity analysis would not fully capture all of the uncertainties in the CO2 field, but this test does reveal where models could show significant differences (particularly because they are not anchored to many observations in the region). To re-do the inversion using the full-physics product would be out of scope of this paper, as that dataset would have to be carefully assessed as well (the full-physics product for example, could be affected by clouds and aerosols). We have instead added a note about the limitations of these tests.

The text has now been updated as follows:

Line [196-198] We re-ran the inversion for each of the ten datasets for the full 2010-2018 time period, which allowed us to investigate random errors in XCO2. However, additional uncertainties could nevertheless remain due to sparse CO2 observations in the region.

4) Line 216 to 220: The September 2010 biomass burning emissions difference of be-tween GFED and the inversion is not a good example of "Our analysis shows that individual years exhibit features that are not present in the bottom-up estimates." As both estimates show 2010 has the highest biomass burning emissions and emission peaks in September.

We have clarified the statement here that, though the September 2010 feature is reflected in the GFED prior, our estimated emissions are still significantly higher than GFED.

The text has now been updated as follows:

Line [221] Our analysis shows that individual years show some differences from the bottom-up estimates.

5) Line 205: The authors write "The inversion results show that the Biomass burning emis-sion rose by $1\pm0.4$ Tg/yr between 2011-2013 to 2014-2018". Can this be checked in the GFED data of more recent years?

The latest GFED product shows an increase between these two periods of 0.4 Tg/yr. It should be noted that the product produced from 2017+ is not the final product (classed as a beta product) and thus there is uncertainty around this exact figure. The presence of a rise of this magnitude seems largely consistent with the result presented in the paper. However, for consistency as we did not use the beta product in our prior, we have not added this discussion to the text.

Technical Corrections:

Line 11: The sentence is difficult to understand. Writing it as an inline list will make it easier to read.

The text has now been updated as follows:

[Lines 12-13] We show that satellite data is beneficial for constraining national-scale

CH4 emissions, and, through a series of sensitivity studies and validation experiments using data not assimilated in the inversion, we demonstrate that (a) calibrated ground-based data are important to include alongside satellite data in a regional inversion, and that (b) inversions must account for any offsets between the two data streams and their representations by models.

Line 220: remove the double "is"

Updated.

Line 345: "modelling but" => "modelling, but"

Updated.

Please also note the supplement to this comment:
https://acp.copernicus.org/preprints/acp-2020-438/acp-2020-438-AC2-supplement.pdf

---

## Author Comment (AC3) · 21 Aug 2020

Thank you for pointing this out - this has now been corrected in all places where this was incorrect within the text.
* * *